# A TonB-dependent transporter is required for secretion of protease PopC across the bacterial outer membrane

Nuria Gómez-Santos [1], Timo Glatter[1], Ralf Koebnik[2], Magdalena Anna Świątek-Połatyńska[1] & Lotte Søgaard-Andersen [1]

TonB-dependent transporters (TBDTs) are ubiquitous outer membrane β-barrel proteins that import nutrients and bacteriocins across the outer membrane in a proton motive force-dependent manner, by directly connecting to the ExbB/ExbD/TonB system in the inner membrane. Here, we show that the TBDT Oar in *Myxococcus xanthus* is required for secretion of a protein, protease PopC, to the extracellular milieu. PopC accumulates in the periplasm before secretion across the outer membrane, and the proton motive force has a role in secretion to the extracellular milieu. Reconstitution experiments in *Escherichia coli* demonstrate that secretion of PopC across the outer membrane not only depends on Oar but also on the ExbB/ExbD/TonB system. Our results indicate that TBDTs and the ExbB/ExbD/TonB system may have roles not only in import processes but also in secretion of proteins.

[1] Max Planck Institute for Terrestrial Microbiology, Karl-von-Frisch Str. 10, 35043 Marburg, Germany. [2] IRD, Cirad, Interactions Plantes Microorganismes Environnement, University of Montpellier, 34394 Montpellier, France. Correspondence and requests for materials should be addressed to L.S.-A. (email: sogaard@mpi-marburg.mpg.de)

Protein secretion is used by all cells to deliver proteins to different cellular compartments. In bacteria, proteins secreted to the extracellular milieu play key roles in a multitude of important processes including virulence, biofilm formation, adhesion, interactions between bacteria in microbiomes, host-microbe interactions, adaptation, and motility. In Gram-negative bacteria, such proteins are synthesized in the cytoplasm and then transported across the inner membrane (IM) as well as the outer membrane (OM). Passage of the two membranes involves either one-step mechanisms directly from the cytoplasm to the extracellular milieu or two-step mechanisms, whereby proteins are first translocated from the cytoplasm across the IM to the periplasm and then from the periplasm across the OM[1]. In two-step mechanisms, proteins are guided by their signal peptides to the Sec or Tat system and then translocated across the IM to the periplasm[2,3]. In parallel, the signal peptide is cleaved off[4]. So far, the final step across the OM has been shown to be mediated either by the type II secretion system (T2SS), the type V secretion system (T5SS), the type IX secretion system (T9SS), or porins[1,5].

In response to starvation, the Gram-negative deltaproteobacterium *Myxococcus xanthus* initiates a multicellular developmental program that culminates in the formation of fruiting bodies inside which the rod-shaped cells differentiate to spores[6]. Fruiting body formation involves two morphogenetic events, aggregation of cells to generate mounds and sporulation. These two events are highly coordinated with aggregation occurring during the first 24–48 h; subsequently, only those cells that have aggregated inside the mounds differentiate to form spores, finally, giving rise to mature fruiting bodies after 72–120 h. The protease PopC (accession number Q1DFT5) is essential for completion of this developmental program[7]. PopC is a subtilisin-like protease and is slowly secreted to the extracellular milieu by starving cells[7]. PopC has a size of 50.8 kDa and is composed of an N-terminal part with no recognizable domains and a C-terminal subtilisin domain[7]. Moreover, sequence analysis previously suggested that PopC does not have a signal peptide[7]. Interestingly, PopC accumulates in non-starving cells as well as in starving cells; however, PopC is only secreted to the extracellular milieu by starving cells[7]. The starvation-induced secretion of PopC depends on the RelA-induced stringent response with accumulation of (p)ppGpp[8,9]. Stringent response by an unknown mechanism results in degradation of PopD, which forms a complex with PopC and inhibits PopC secretion in non-starving cells[8]. Once secreted, PopC directly cleaves the cell surface-exposed p25 protein[7,10], which is encoded by the *csgA* gene, to generate the cell surface-exposed p17 protein[7]. p17 is often referred to as the intercellular C-signal and is essential for fruiting body formation[11,12]. Intercellular C-signal transmission has been suggested to depend on direct cell-cell contacts involving pole-to-pole contacts between the rod-shaped *M. xanthus* cells[13]. In the current model, the C-signal induces aggregation and sporulation at distinct thresholds[14–16]. The slow, regulated accumulation of p17 during starvation[15] together with the contact-dependent signaling mechanism has been suggested to ensure the precise temporal and spatial coordination of aggregation and sporulation[15,17]. Additionally, it has been suggested that the slow secretion of PopC to the extracellular milieu contributes to the slow accumulation of p17[7].

To begin to understand how PopC secretion is regulated in response to the nutritional status of cells, we focused on elucidating how PopC is secreted to the extracellular milieu. Here we show that PopC is secreted as a full-length protein in a two-step mechanism. Moreover, we show that PopC secretion across the OM depends on a TonB-dependent transporter (TBDT) in the OM together with a functional ExbB/ExbD/TonB system (henceforth, Ton system) in the IM. TBDTs form 22-stranded β-barrels in the OM with a plug domain that occludes the lumen of the β-barrel[18]. TBDTs have previously been shown to mediate the import of nutrients such as carbohydrates, vitamin B12, iron complexes, and nickel chelates as well as bacteriocins, which are up to 74 kDa in size[19], across the OM to the periplasm[18]. These import processes depend on the proton motive force (PMF) across the IM[18,20]. To this end, TBDTs in the OM connect to the Ton system in the IM[18–20]. The Ton system harness the PMF across the IM and energize TBDTs via a direct interaction between the periplasmic C-terminal domain (CTD) of TonB[18,20] and a short conserved sequence motif, the so-called TonB box, in the N-terminal plug domain of the TBDTs[21–23]. Our results indicate that TBDTs and the ExbB/ExbD/TonB system may have roles not only in import processes but also in secretion of proteins.

## Results

**PopC is secreted as a full-length protein.** To determine whether the presence of PopC in the cell-free supernatant is the result of secretion or the release of OM vesicles (OMV), we isolated OMV from cells of the wild-type (WT) strain DK101 starved for 6 h. While PopC was detected in the cell-free supernatant, it was not detected in OMV; by contrast, the control protein Oar, which is an OM protein[24] that is also present in OMV[24,25], was only detected in OMV (Supplementary Fig. 1a).

To determine whether PopC is secreted to the extracellular milieu in a processed form or as a full-length protein, we isolated total cell extract as well as the cell-free supernatant from WT *M. xanthus* cells starved for 6 h, and determined whether PopC from the cell-free supernatant contains the N terminus of native PopC. The native PopC N terminus was detected in both fractions (Supplementary Fig. 1b). In total, we conclude that PopC is secreted to the extracellular milieu as a full-length protein.

**PopC is enriched in the periplasm before secretion.** PopC was previously suggested to localize to the cytoplasm prior to secretion to the extracellular milieu because sequence analysis indicated that it does not contain a signal peptide[7]. To determine experimentally to which subcellular compartment PopC localizes before secretion to the extracellular milieu, cell lysates of non-starving and starving *M. xanthus* cells were fractionated into four fractions enriched for proteins in the cell-free supernatant, cytoplasmic proteins, periplasmic proteins, or membrane proteins. Control proteins previously shown to localize to these compartments documented that the fractionation procedure worked properly. In these experiments, PopC was strongly enriched in the periplasmic fraction in non-starving as well as in starving cells (Fig. 1a, b; left and middle panels).

We previously showed that starving cells, in the presence of the translation inhibitor chloramphenicol, secrete PopC to the extracellular milieu following the same kinetics as in untreated starving cells during the first 24 h of starvation, i.e. the PopC level in total cell extracts slowly decreases while the level in the cell-free supernatant slowly increases[8]. Thus, it was concluded that PopC synthesis and secretion are not coupled and that preformed PopC is secreted to the extracellular milieu[8]. Therefore, to determine whether PopC is secreted to the extracellular milieu from the cytoplasm or the periplasm, we examined the level of PopC accumulation in fractions enriched for periplasmic and cytoplasmic proteins as well as in the cell-free supernatant in starving cells treated with chloramphenicol. Over the course of the experiment, the PopC level in the periplasm slowly decreased while the level in the cytoplasm did not decrease (Supplementary Fig. 1c). Moreover, and as previously observed[8], the PopC level in the cell-free supernatant increased (Supplementary Fig. 1c). In total, these

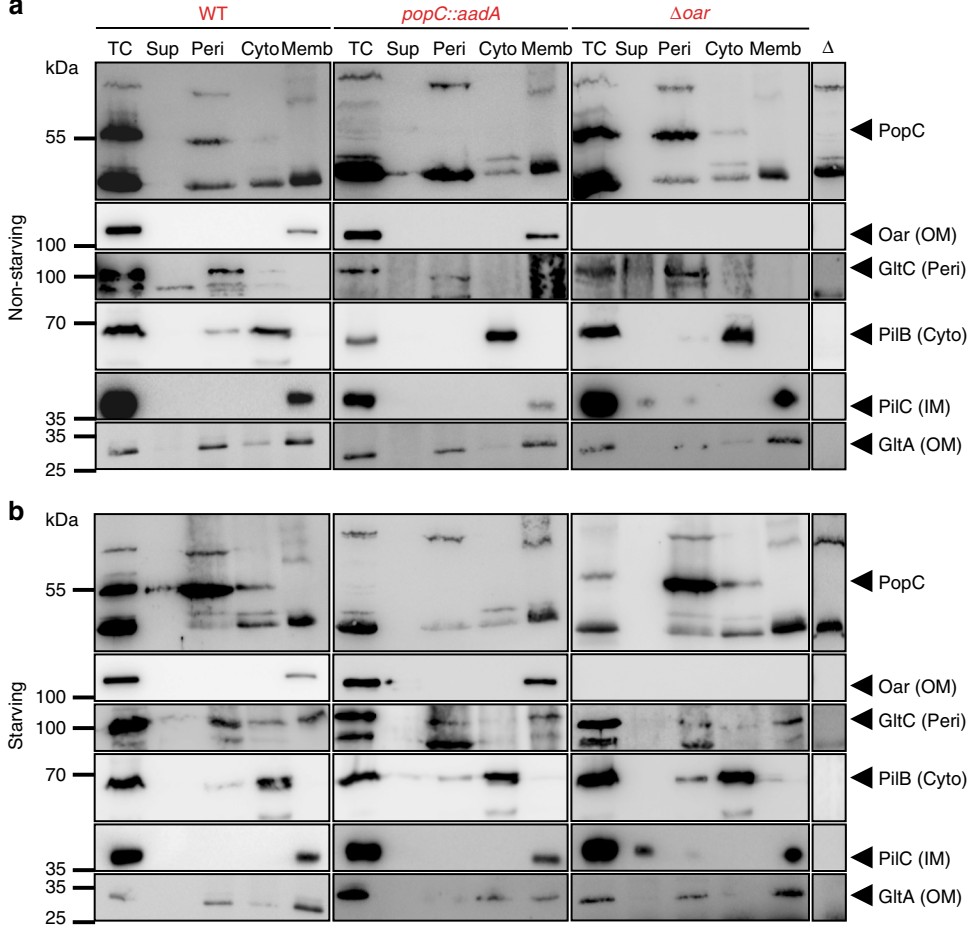

**Fig. 1** PopC is enriched in the periplasm before secretion to the extracellular milieu. **a**, **b** Immunoblots using antibodies against the indicated proteins in total cell extracts (TC) from **a** non-starving and **b** 6 h starving cells of the indicated genotypes (red) fractionated into cell-free supernatant (Sup), periplasm (Peri), cytoplasm (Cyto), and membranes (Memb). Oar, GltC, PilB, PilC, and GltA, and antibodies against these proteins serve as markers for the indicated fractions. Lanes labeled with an open triangle: TC of individual in-frame deletion mutants corresponding to protein tested. Note that throughout the text, the popC::aadA allele is referred to as popC. Source data are provided as a Source Data file

observations demonstrate that PopC is highly enriched in the periplasm. Moreover, they support that PopC is secreted to the extracellular milieu from the periplasm supporting that PopC secretion to the extracellular milieu occurs in a two-step mechanism.

**PMF has a role in PopC secretion**. To determine whether PopC secretion across the OM is energy-dependent, we quantitatively determined PopC secretion in starving WT cells treated with compounds that dissipate or reduce PMF (carbonyl cyanide m-chlorophenyl hydrazine (CCCP)), pH gradient (nigericin), membrane potential (valinomycin), or cellular ATP content (arsenate). In M. xanthus, 10 μM CCCP and 100 μM nigericin have been reported to dissipate the PMF and the pH gradient, respectively[26]. However, in our hands, concentrations higher than 5 μM CCCP and 50 μM nigericin caused cell lysis. Therefore, we treated cells with 5 μM CCCP and 50 μM nigericin and, consequently, the PMF and pH gradient may not be completely dissipated. Following addition of the different compounds, the cell-free supernatants were analyzed to determine the cumulative level of PopC. CCCP and nigericin significantly and reversibly reduced PopC secretion within 30 min, and PopC levels did not change significantly after 10 min of exposure to CCCP and nigericin while valinomycin and arsenate had no significant effect on PopC

secretion (Fig. 2). Importantly, ATP content only decreased significantly in response to arsenate (Supplementary Fig. 2). While it cannot be excluded that CCCP and nigericin indirectly affect PopC secretion, these data support the notion that the PMF and the pH gradient might have a role in PopC secretion across the OM.

The M. xanthus genome encodes a T2SS[6], no T5SS[6], no T9SS[27], and several porins[24]. Attempts to delete the genes for the T2SS were unsuccessful; however, only a few T2SS substrates have been shown to depend on PMF for secretion across the OM[28,29] and the majority of substrates secreted by T2SS depends on ATP hydrolysis[1] suggesting that the T2SS is not involved in PopC secretion. Similarly, OM porins function independently of ATP and PMF[30]. Altogether, these observations suggested that PopC could be secreted from the periplasm across the OM by an unidentified mechanism that might involve the PMF.

**The TBDT Oar is required for PopC secretion**. PMF-dependent processes in the OM include those involving TBDTs in the OM and a Ton system composed of the IM proteins TonB, ExbB, and ExbD[18,20]. Because TBDTs are energized by the PMF and directly involved in translocation processes across the OM including that of large bacteriocin molecules, they were candidates for being involved in PopC secretion across the OM. The M. xanthus

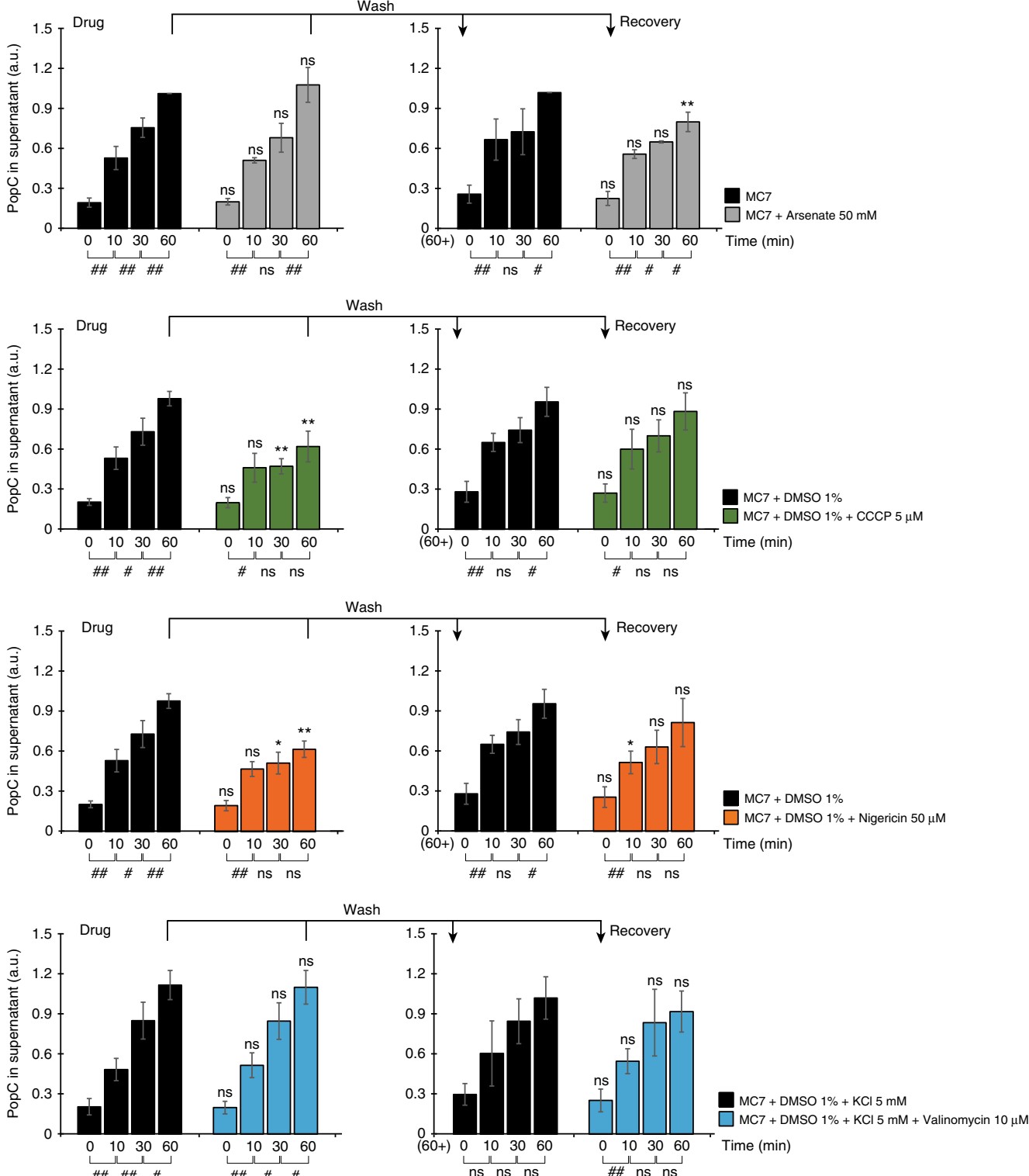

**Fig. 2** PMF has a role in PopC secretion from the periplasm across the outer membrane. WT cells were simultaneously exposed to starvation and the indicated drugs. After 60 min, drugs were washed away. Cell-free supernatants (Sup) were collected at indicated time points and analyzed by ELISA using α-PopC antibodies. For each condition, the PopC signal detected after 60 min in the untreated sample (Drug and Recovery) was used to normalize the remaining values. $n = 4$. Error bars: s.d. Two comparisons were done to determine the effect of a drug on PopC secretion. First, treated samples were compared to the untreated sample from the same time point by a $t$-test, \*$p$-value $\leq 0.05$; \*\*$p$-value $\leq 0.01$; and ns, $p$-value $> 0.05$. Second, to detect significant changes in PopC accumulation in Sup along an experiment, samples at each time point were compared to the samples from the previous time point for each individual condition by a $t$-test, #$p$-value $\leq 0.05$; ##$p \leq 0.01$; and ns, $p$-value $> 0.05$, respectively (shown below the time line). Note that in this comparison, $p > 0.05$ indicates that secretion is blocked. Data for MC7 + DMSO in the diagrams for CCCP and nigericin are the same. Source data are provided as a Source Data file

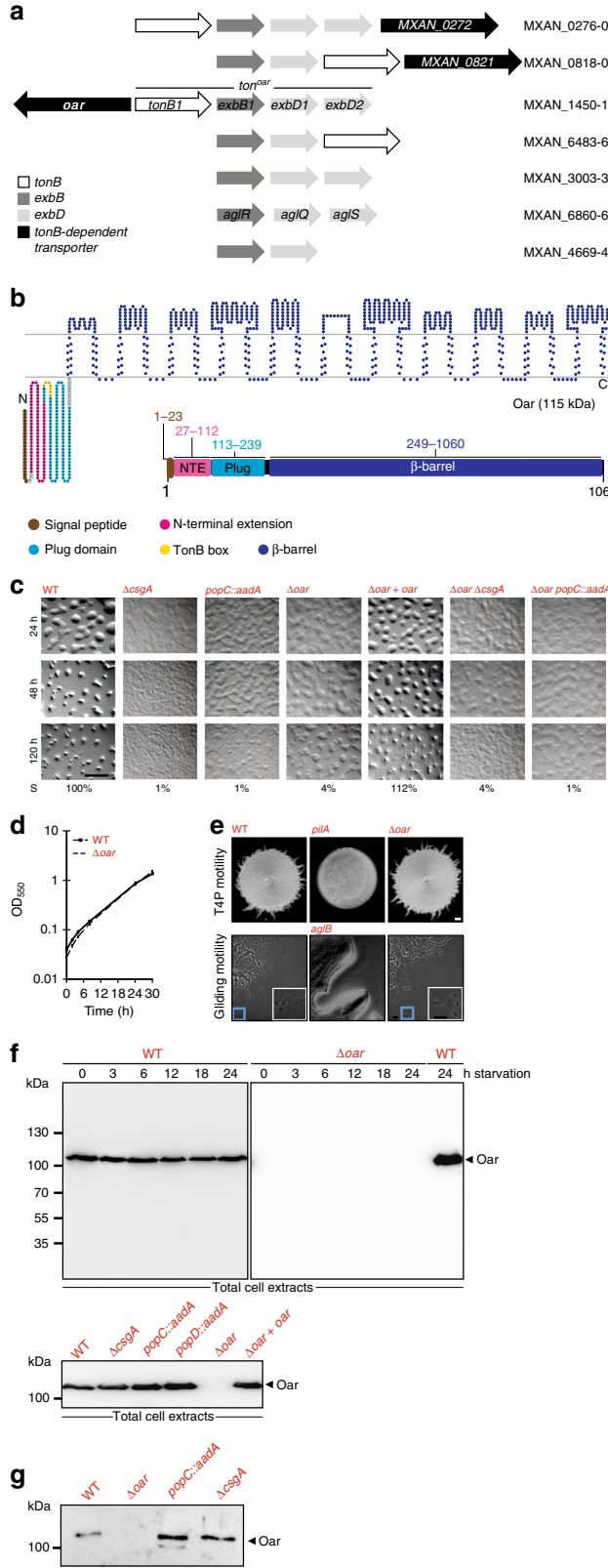

**Fig. 3** Oar is a TBDT important for development. **a** Gene clusters in *M. xanthus* encoding ExbB and ExbD homologs. Arrows indicate direction of transcription. MXAN numbers indicate the first and the last genes in each cluster. The four genes encoding the Ton system associated with Oar (*ton^oar*) are indicated. **b** Predicted membrane topology and domain structure of Oar. **c** Phenotype of strains of indicated genotypes during development. Strains were imaged at indicated time points of starvation. Sporulation (S) is expressed as percentage of WT. Scale bar, 1 mm. **d** Growth of WT and Δ*oar* mutant. **e** Motility of Δ*oar* mutant. Large inserts show areas indicated by blue boxes at a higher magnification. The *pilA* mutant that lacks type IV pili and the *aglB* mutant that lacks gliding motility were used as negative controls. Scale bars: 10 μm. **f** Immunoblot detection of Oar using α-Oar antibodies in total cell extracts of non-starving (0 h) and starving WT and Δ*oar* cells (upper panel) and non-starving cells of the indicated genotypes (lower panel). **g** Immunoblot detection of Oar in OMV in strains of the indicated genotypes under non-starving conditions. Source data for **f** and **g** are provided as a Source Data file

MXAN_0272 affected neither development nor PopC secretion (Supplementary Fig. 3a, b). Attempts to generate an in-frame deletion of MXAN_0821 were unsuccessful indicating that this TBDT is essential. In agreement with previous observations[31,24], lack of the TBDT MXAN_1450, which is also referred to as Oar[31], caused impaired fruiting body formation and sporulation (Fig. 3b, c). Importantly, Oar has previously been implicated in synthesis or export of the intercellular C-signal[24]. Therefore, we focused on the analysis of Oar in PopC secretion.

The Δ*oar* mutant had a growth rate similar to WT (Fig. 3d) and displayed normal type IV pili-dependent motility and gliding motility (Fig. 3e) both of which are important for development[32]. Moreover, Oar accumulated equally in non-starving and starving cells (Fig. 3f; upper). Finally, in a Δ*oar* strain in which *oar* was ectopically expressed from the *pilA* promoter on a plasmid integrated in the genome at the *attB* site, Oar accumulated at the same level as in WT (Fig. 3f, lower) and complemented the developmental defects caused by the Δ*oar* mutation (Fig. 3c). We conclude that Oar is essential for development.

As discussed above, Oar is an OM protein[24] that is also present in OMV[24,25] (see also Supplementary Fig. 1a). Oar is 300–400 amino acids longer than typical TBDTs (Supplementary Table 1). Nevertheless, sequence analysis supports that Oar contains a β-barrel domain characteristic of TBDT (Pfam domain PF00593; TonB_dep_Rec) and that Oar forms a 22-stranded β-barrel with a topology similar to that of structurally characterized TBDTs (Fig. 3b). Also, Phyre2 predictions support that Oar has a structure similar to that of structurally characterized TBDTs (Supplementary Table 1) forming a 22-stranded β-barrel with a diameter of ≈35–40 Å. Moreover, Oar is predicted to contain the N-terminal plug domain (Pfam domain PF07715; Plug) including a TonB box characteristic of TBDTs (Fig. 3b and Supplementary Fig. 4). In addition to the plug domain, Oar contains an N-terminal extension (NTE) with the Pfam domain PF13620 (CarboxypepD_reg) (Fig. 3b), which has previously been reported to be present in the N terminus of the Oar subclass of TBDTs[33]. This domain is distinct from the N-terminal domain (Pfam domain PF07660; STN) found in TBDTs involved in signal transduction[33] (see also below).

To determine whether *oar* acts in the same genetic pathway as *popC* and *csgA*, we performed genetic epistasis experiments in which the developmental phenotype of the three single mutants was compared to that of the two double mutants using fruiting body formation and sporulation as phenotypic readouts. The Δ*oar popC* mutant and the Δ*oar* Δ*csgA* mutant had the same phenotype as the *popC* mutant and the Δ*csgA* mutant,

genome contains seven gene clusters encoding ExbB and ExbD homologs[26] (Fig. 3a). Four of these gene clusters also encode a TonB homolog; and, among these four clusters, three encode a TBDT. Here we focused on the three gene clusters encoding a TBDT and a complete Ton system. Lack of the TBDT encoded by

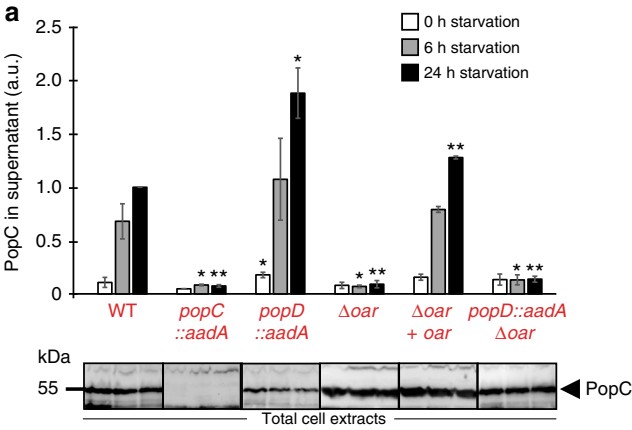

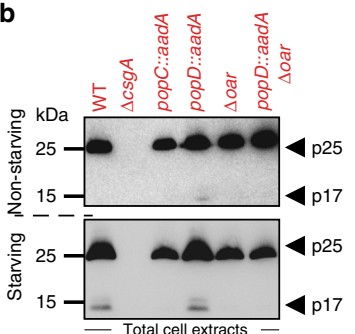

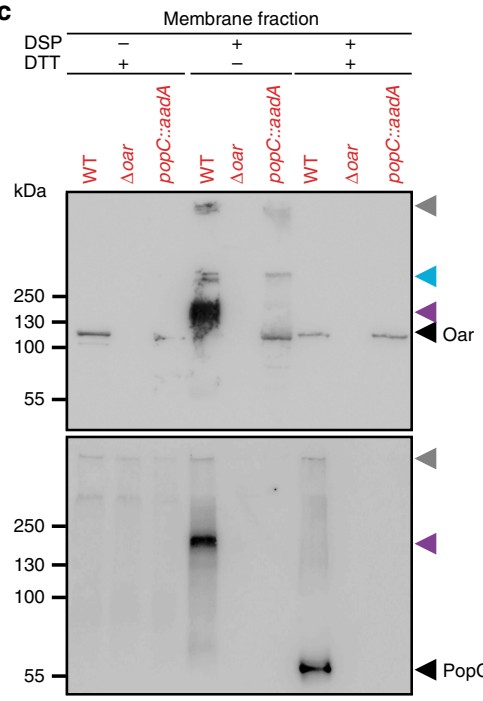

**Fig. 4** Oar is essential for PopC secretion. **a** PopC secretion depends on Oar. Upper panel, cell-free supernatants from strains of indicated genotypes were analyzed by ELISA using α-PopC antibodies. The signal detected in WT after 24 h was used to normalize the remaining values. 1 a.u. corresponds to 3 ± 1% of PopC in total cell extracts. $n = 3$. Error bars: s.d. Each sample from a mutant was compared to WT sample from the same time point by $t$-test, *$p$-value $\leq 0.05$, **$p$-value $\leq 0.01$, respectively. Lower panel, immunoblots using α-PopC antibodies of PopC accumulation in total cell extracts during starvation in the indicated strains. Note that throughout the text, the popD::aadA allele is referred to as popD. **b** p17 is not generated in Δoar mutant. Immunoblots using α-p25/p17 antibodies of p25 and p17 accumulation in total cell extracts from non-starving and 6 h starving cells of indicated genotypes. **c** Oar-dependent PopC membrane association. Six-hour starving cells of the indicated genotypes were exposed to DSP, fractionated, and treated with DTT as indicated. Oar (upper) and PopC (lower) were detected by immunoblot using α-PopC and α-Oar antibodies. Black arrowheads, monomeric Oar or PopC; gray arrowheads, loading wells; blue arrowhead, crosslinked Oar product observed in WT and in the absence of PopC; purple arrowheads, crosslinked Oar-PopC product. Source data are provided as a Source Data file

mutant (Fig. 4a; lower). Importantly, the Δoar mutant was impaired in PopC secretion (Fig. 4a; upper). Also, and as predicted based on the PopC secretion defect, the Δoar mutant did not accumulate p17, whereas p25 accumulation was as in WT (Fig. 4b). As expected, ectopic expression of oar from the pilA promoter in the Δoar mutant complemented the defects in PopC secretion (Fig. 4a). Based on these results, we conclude that Oar is essential for PopC secretion.

**Oar interacts with PopC and has a role in its secretion.** PopC secretion in response to starvation depends on the stringent response. To begin to pinpoint the function of Oar in PopC secretion, we asked whether the stringent response is affected in the Δoar mutant. To this end, we determined ppGpp accumulation in WT and the Δoar mutant in response to starvation. In these experiments, Δoar mutant accumulated ppGpp similarly to WT (Supplementary Fig. 5).

Stringent response has been suggested to result in the degradation of PopD, which inhibits PopC secretion in non-starving cells[8]. To determine whether Oar functions upstream or downstream of PopD in PopC secretion, we determined PopC secretion in a popD Δoar double mutant. As previously shown[8], the popD mutant secreted PopC at a slightly but significantly higher level than WT (Fig. 4a; upper) and accumulated Oar at WT levels (Fig. 3f; lower). However, neither PopC secretion nor p17 formation was detected in the popD Δoar double mutant while this mutant still accumulated WT levels of PopC and p25 in total cell extracts (Fig. 4a; lower, b). In total, these observations suggest that Oar functions downstream of PopD and support a model in which Oar may be acting at the level of PopC secretion.

If Oar is directly involved in PopC secretion across the OM, the prediction is that the two proteins interact directly. To test for direct interaction between Oar and PopC, we used a dithiobis (succinimidyl propionate) (DSP) in vivo crosslinking approach in which starving M. xanthus cells of the WT, the Δoar mutant and the popC mutant were treated with DSP followed by isolation of the membrane fraction and breaking of crosslinks with dithiothreitol (DTT). In the absence of crosslinking, Oar but not PopC associated with the membrane fraction (Fig. 4c; see also Fig. 1a, b). However, the membrane fraction from crosslinked WT contained a high-molecular weight crosslink that was detected by α-PopC antibodies (Fig. 4c; upper) as well as by

respectively (Fig. 3c) demonstrating that the three genes act in the same genetic pathway. Of note, the ΔcsgA and popC mutants accumulated Oar at WT levels in total cell extracts (Fig. 3f; lower) as well as in the OM (Fig. 3g).

Next, we tested whether Oar is important for PopC secretion to the extracellular milieu. First, we observed that PopC is enriched in the periplasm in non-starving as well as in starving cells of the Δoar mutant (Fig. 1a, b). Furthermore, PopC accumulated at WT levels in total cell extracts under both conditions in the Δoar

α-Oar antibodies (Fig. 4c; lower), and this crosslink was observed neither in the Δ*oar* mutant nor in the *popC* mutant (Fig. 4c). Upon DTT treatment of the crosslinked membrane fractions, PopC that migrated as a monomer was recovered from the membrane fraction of WT but not from the membrane fraction of the Δ*oar* mutant (Fig. 4c; lower). These observations support the notion that PopC and Oar interact directly and that Oar is directly involved in secretion of PopC. The observation that PopC is only detected in the membrane fraction after crosslinking (compare Fig. 1a, b and 4c; lower) also suggests that the two proteins may only interact transiently.

**TonB box in the Oar plug domain is important for PopC secretion**. To test if Oar depends on a functional Ton system to support PopC secretion, we used bacterial two-hybrid (BACTH) analysis to determine whether the N-terminal part of Oar including the plug domain and the NTE interacts with the peri-plasmic CTD of TonB1 of the Oar-related Ton system (henceforth, Ton$^{Oar}$ system) (Fig. 3a). As shown in Fig. 5a, b, the N-terminal part of Oar interacts with the CTD of TonB1 supporting a direct connection between Oar and the Ton$^{Oar}$ system.

To study the function of the N-terminal domains of Oar in PopC secretion, we created strains harboring Oar variants that lacked the plug domain, the NTE, or both the plug domain and the NTE (Fig. 5c). The Oar variant lacking the NTE accumulated in total cell extracts as well as in the OM at WT levels (Fig. 5c, d) and supported PopC secretion (Fig. 5c). The Oar variant that lacked the plug domain accumulated at a lower level than the WT protein in total cell extracts and appeared to be degraded during starvation, but was still present in the OM (Fig. 5c, d). This variant did not support PopC secretion to the extracellular milieu (Fig. 5c). Finally, an Oar variant lacking both plug and NTE did not accumulate and, as expected, the mutant did not secrete PopC (Fig. 5c, d).

To try to further clarify whether the Oar plug domain has a function in PopC secretion, we focused on the conserved TonB box in the plug domain. In well-characterized TBDTs, the TonB box mediates the interaction between the plug domain and the periplasmic CTD of TonB. Therefore, if the Oar plug domain is important for PopC secretion, then the prediction is that the TonB box is also important for PopC secretion. To test this prediction, we generated two Oar variants with substitutions in conserved residues in the TonB box of Oar that reduce or block the activity in well-characterized TBDTs[34] (Fig. 5c and Supplementary Fig. 4). We observed that while the two Oar variants accumulated as the WT protein in total cell extracts and the OM (Fig. 5c, d), they were both significantly reduced in supporting PopC secretion (Fig. 5c). We conclude that the TonB box in the Oar plug domain is important for Oar function in PopC secretion, likely by directly interacting with the CTD of TonB1.

**Analysis of Ton$^{Oar}$ system in PopC secretion**. To establish a direct functional link between Oar and the Ton$^{Oar}$ system (Fig. 3a), we systematically generated in-frame deletions in each of the four genes encoding the proteins of this system. Analyses of these mutants demonstrated that after 120 h of starvation the Δ*exbD2*, Δ*exbD1*, and Δ*exbB1* mutants had a developmental phenotype similar to that of the Δ*oar* mutant, whereas the Δ*tonB1* mutant had a more severe defect in fruiting body formation (Fig. 6a). PopC accumulated in each of the four *ton$^{oar}$* system mutants. Surprisingly, the Δ*exbD2*, Δ*exbD1*, and Δ*exbB1* mutants still secreted PopC albeit at a significantly lower level than in WT (Fig. 6b); by contrast, the Δ*tonB1* mutant was essentially blocked in PopC secretion (Fig. 6b). The developmental defects of the Δ*exbD2*, Δ*exbD1*, and Δ*exbB1* mutants were

complemented by ectopic expression of the relevant WT gene from the *pilA* promoter, whereas the Δ*tonB1* mutant was not complemented by ectopic expression of *tonB1* (Fig. 6a). Of note, the Δ*exbD2*, Δ*exbD1*, and Δ*exbB1* mutants accumulated Oar, whereas the Δ*tonB1* mutant did not (Fig. 6c; upper) suggesting that the Δ*tonB1* mutation affected either *oar* expression or Oar accumulation. To distinguish between these two possibilities, *tonB1* or *oar* was expressed ectopically in the Δ*tonB1* mutant from the *pilA* promoter. Upon ectopic expression of *oar* in the Δ*tonB1* mutant, Oar accumulated (Fig. 6c; lower) without restoring the developmental defects (Fig. 6a). However, ectopic expression of *tonB1* did not restore Oar accumulation (Fig. 6c; lower). The distance between the start codon of *oar* and the 5′-end point of the *tonB1* deletion is 249 bp. Therefore, these observations suggest that the Δ*tonB1* mutation caused a reduction in *oar* expression and that TonB1 is not important for Oar accumulation. Consistently, we observed using quantitative real-time-PCR (qRT-PCR) on total RNA isolated from non-starving cells that *oar* was expressed in WT but not in the Δ*tonB1* mutant (Fig. 6d). Moreover, these observations suggest that the reason why the Δ*tonB1* mutant cannot be complemented by ectopic expression of *tonB1* is lack of *oar* expression.

For subsequent analyses, we generated a Δ*tonB1-exbD2* quadruple mutant (Δ*ton$^{oar}$* mutant). This mutant has the same 5′-end point in *tonB1* as the Δ*tonB1* mutation. Nevertheless, the Δ*ton$^{oar}$* mutant expressed *oar* (Fig. 6d) and accumulated Oar (Fig. 6c; upper) in the OM (Fig. 6e). We speculate that the defect in *oar* expression in the single Δ*tonB1* mutant is overcome by a heterologous promoter region downstream of the complete *ton$^{oar}$* system when all four genes are deleted, allowing *oar* expression and, consequently, Oar accumulation. The Δ*ton$^{oar}$* mutant had developmental defects (Fig. 6a), accumulated PopC in total cell extracts and secreted PopC in an Oar-dependent manner at WT levels (Fig. 6b). Surprisingly, however, the Δ*ton$^{oar}$* mutant did not accumulate p17 while p25 accumulated at WT levels (Fig. 6f).

To reconcile that the Δ*ton$^{oar}$* mutant still secreted PopC but the Oar plug domain with an intact TonB box is important for PopC secretion and that the N-terminal domains of Oar interact with the CTD of TonB1, we hypothesized that in the absence of the Ton$^{Oar}$ system, Oar could be energized by one of the remaining Ton systems in *M. xanthus* (Fig. 3a). Consistent with this idea, the N-terminal domains of Oar including the plug domain with the TonB box interact with the CTD of the TonB proteins MXAN_0276 and MXAN_0820 in BACTH analysis (Fig. 3a and Supplementary Fig. 6).

To reconcile that the Δ*ton$^{oar}$* mutant secreted PopC but did not cleave p25 to generate p17, we considered that previous work had suggested that intercellular C-signal transmission depends on direct pole-to-pole contacts between the rod-shaped *M. xanthus* cells[13] raising the possibility that the Ton$^{Oar}$ system could be involved in regulation of the subcellular localization of p17/p25 and/or Oar. Therefore, we determined the localization of p17/p25 and Oar in starving cells by immunofluorescence microscopy. In these experiments, we observed that p17/p25 mostly localized polarly in starving cells and that this localization was independent of Oar and the Ton$^{Oar}$ system (Fig. 6g; upper). Importantly, Oar also mostly localized polarly in starving WT cells but this polar localization depended on the Ton$^{Oar}$ system (Fig. 6g; lower). Thus, the Ton$^{Oar}$ system ensures polar localization of Oar (Fig. 6h). We speculate that in the absence of the Ton$^{Oar}$ system, Oar is energized by one of the remaining Ton systems; however, in this situation, PopC would be secreted away from its substrate (Fig. 6h). Because PopC has a short half-life after secretion to the extracellular milieu[7], PopC secreted away from its substrate may not efficiently cleave p25 and, thus, p17 would not accumulate.

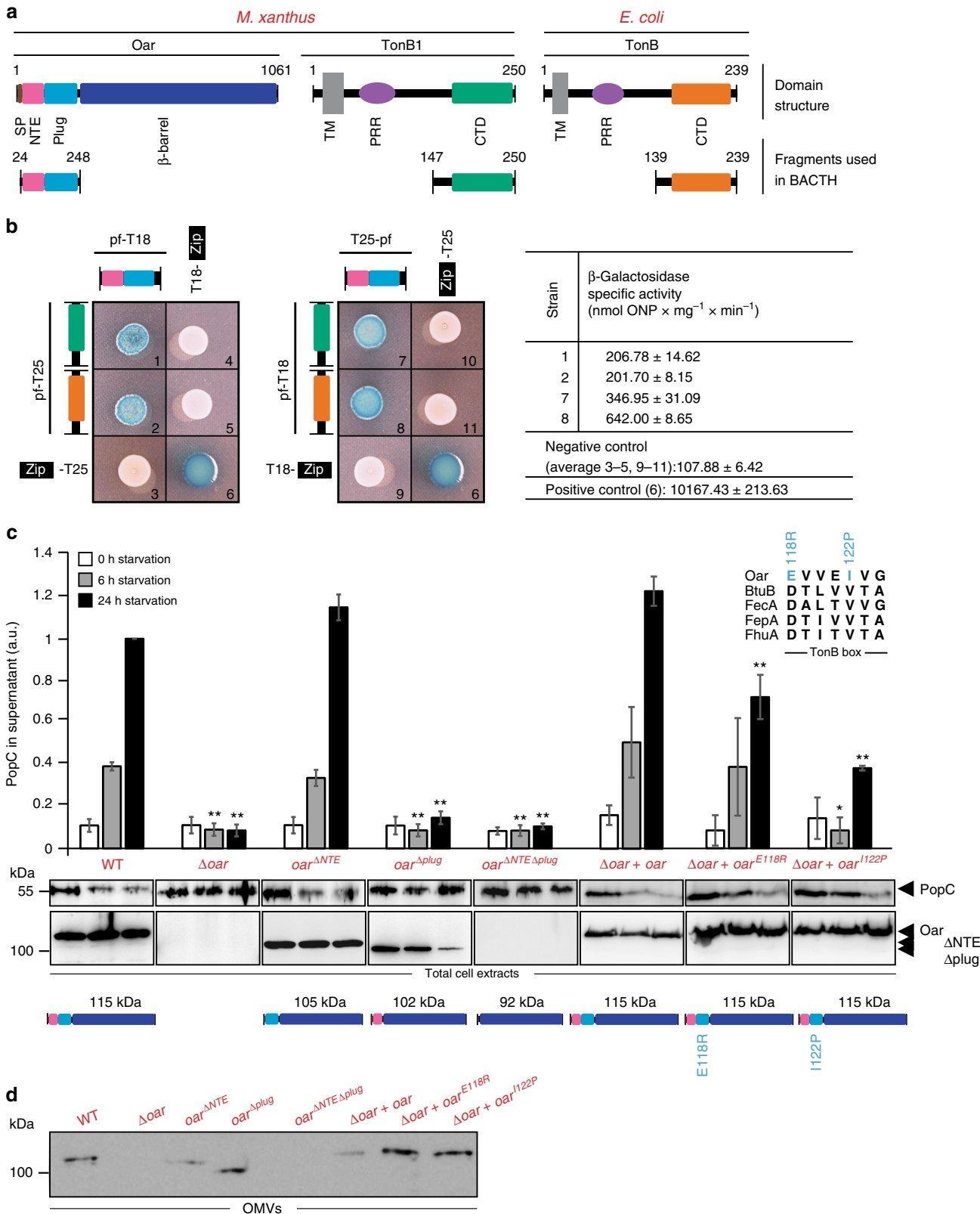

**PopC secretion by *Escherichia coli* depends on Oar and the Ton system**. The presence of several Ton systems in *M. xanthus* complicates the analysis of the functional connection between Oar and the Ton[Oar] system in PopC secretion. Therefore, we used heterologous expression experiments in *E. coli*, which only contains a single Ton system that energizes several TBDTs[20,35]. To

this end, we first established that the N-terminal region of Oar with the plug domain and the NTE, interacted with the CTD of TonB from *E. coli* in BACTH analyses (Fig. 5a, b).

Subsequently, in experiments in which native Oar and/or PopC were expressed in WT *E. coli*, we observed that PopC solubility increased when co-expressed with Oar but not when co-expressed

**Fig. 5** TonB box in Oar plug domain is important for Oar function. **a** Domain structure of Oar, TonB1, and TonB, and fragments used in BACTH analysis. SP signal peptide, NTE N-terminal extension, TM transmembrane domain, PRR proline-rich region, CTD TonB C-terminal domain. **b** BACTH analysis for interactions between N-terminal domains of Oar and the CTD of TonB1 or TonB. The indicated protein fragments (pf) were fused to the N or C terminus of the T25 and T18 fragments of CyaA. Oar, TonB1, and TonB domains are shown as in **a**. Zip indicates that the leucine zipper from GCN4 fused to T25 and T18 and was used as a positive control. Left panels, representative images of *E. coli* strain BTH101 expressing the indicated protein fusions and labeled with numbers. The specific activity of β-galactosidase is shown on the right as mean ± s.d. (*n* = 3) (same strain numbers as in the left panel). **c** Accumulation of PopC during starvation in cell-free supernatants and PopC and Oar in total cell extracts of strains producing Oar variants. Oar variants are shown following the same scheme as in **a**. Inset, TonB box alignment extracted from alignment of plug domain of Oar and TBDTs from *E. coli* (Supplementary Fig. 4); substituted residues in the Oar TonB box are indicated in blue. In the *oar*$^{ΔNTE}$, *oar*$^{Δplug}$, and *oar*$^{ΔNTEΔplug}$ strains, Oar variants are expressed from the native site and these strains are compared to the WT strain; in the Δ*oar* + *oar*, Δ*oar* + *oar*$^{E118R}$, and Δ*oar* + *oar*$^{I122P}$ strains, Oar variants are expressed from the *pilA* promoter and integrated at the *attB* site and these strains are compared. Oar and PopC accumulation was detected and analyzed as in Fig. 1a and Fig. 4a, respectively. *n* = 3. Error bars: s.d. **d** Detection of Oar variants in OMV in strains of the indicated genotypes under non-starving conditions. Source data for **c** and **d** are provided as a Source Data file

with the TBDT MXAN_0272 (Figs. 3a, 7a) supporting that Oar and PopC interact directly. In the strain co-expressing PopC and Oar, Oar was enriched in the membrane fraction (Fig. 7b) and PopC was enriched in the periplasm and, as opposed to PopC in *M. xanthus*, also in the membrane fraction (Fig. 7b). In order to assess in which membrane Oar and PopC were enriched, we separated the IM and OM by sucrose density gradient centrifugation. After separation of the IM and OM, Oar as well as PopC were observed to be enriched in the OM when compared to the IM and OM control proteins TatC and OmpA, respectively (Fig. 7c; upper). Importantly, when expressed in the absence of Oar, PopC did not fractionate with the OM and behaved similarly to the soluble proteins SurA and RpoD (Fig. 7c; middle). In an *E. coli* Δ*tonB* mutant, co-expressed Oar and PopC behaved as in the WT TonB$^+$ strain except that more of PopC associated with the IM than in WT (Fig. 7b, c). We speculate that the differences observed in PopC association with the OM in *E. coli* compared to *M. xanthus* are caused by higher levels of PopC and Oar accumulation in *E. coli* leading to saturation of the Ton system in *E. coli* and detection of translocation intermediates. Remarkably, PopC was exclusively detected in cell-free supernatants when co-expressed with Oar in the WT TonB$^+$ *E. coli* strain but not in the absence of TonB, ExbB, or ExbD (Fig. 7d and Supplementary Fig. 7a, b). Control experiments with proteins that localize in the periplasm, cytoplasm, IM, or OM verified that the presence of PopC in the cell-free supernatant of the WT *E. coli* strain co-expressing Oar and PopC was not the result of cell lysis (Fig. 7d and Supplementary Fig. 7a, b). To exclude the possibility that the presence of PopC in the cell-free supernatant was an artifact caused by its high level of accumulation, we expressed in the WT *E. coli* strain two variants of the MalE protein, cMalE and pMalE that are targeted to the cytoplasm and periplasm, respectively. While both MalE variants were overexpressed, none of them accumulated in the cell-free supernatant (Supplementary Fig. 7c-e). We conclude that the presence of PopC in the cell-free supernatant of the WT *E. coli* strain with an intact Ton system and co-expressing Oar and PopC is the result of bona fide secretion of PopC.

**PopC and Oar interact in *E. coli*.** To determine whether PopC and Oar interact directly in *E. coli*, we performed co-immunoprecipitation (co-IP) experiments with α-PopC antibodies on *E. coli* membranes isolated from WT and the Δ*tonB* mutant expressing PopC and/or Oar. By immunoblotting, we observed that PopC was immunoprecipitated independently of Oar (Fig. 8a; upper), whereas Oar was only immunoprecipitated by the α-PopC antibodies in the presence of PopC (Fig. 8a; lower) supporting that PopC and Oar interact. To more precisely quantify the co-IP experiments, we used label-free quantitative mass spectrometry (LFQ-MS) and observed that in the presence

of PopC, Oar was enriched on average 4.9-fold in WT and 15.1-fold in the Δ*tonB* mutant (Fig. 8b). These observations support that PopC and Oar interact and that Oar is directly involved in secreting PopC.

**Distribution and domain structure of TBDTs.** The data presented show that a TBDT has a role in protein secretion across the OM. To assess how widespread this secretion mechanism could potentially be, we searched the Pfam database[36] for TBDTs and tallied the taxonomic distribution of these proteins (Methods). Our search identified 34,893 TBDTs, 34,743 of which could be assigned to 25 phyla including didermic *Negativicutes* in the Firmicutes (Fig. 9). While all TBDTs per definition contain the TBDT β-barrel domain (PF00593) and the plug domain (PF07715), the N terminus of these proteins have five different domain architectures. The two dominant domain architectures comprise TBDTs without additional N-terminal domains (68%) and the Oar domain architecture (28%). Interestingly, the distribution of TBDT with different N-terminal domain architectures varies significantly between phyla, e.g. TBDTs with a domain architecture similar to that of Oar dominate in Bacteroidetes, whereas TBDTs without additional domains in the N terminus dominate in Proteobacteria.

**Discussion**
In bacteria, proteins secreted to the extracellular milieu are critical for a multitude of important processes. In Gram-negative bacteria, secretion of such proteins represents a formidable challenge because these proteins have to cross two membranes. Gram-negative bacteria have evolved two fundamentally different types of systems that enable the secretion of proteins to the extracellular milieu. One type supports the secretion of a protein in a one-step mechanism directly from the cytoplasm while the second type involves a two-step mechanism in which a protein is first translocated across the IM to the periplasm and then across the OM. In this work, we show that translocation of PopC to the extracellular milieu occurs in a two-step process and that translocation across the OM involves a member of the widespread family of TBDTs in the OM together with a functional Ton system in the IM, in both *M. xanthus* and *E. coli*. Below we discuss the three lines of experimental evidence leading to this conclusion.

First, we demonstrate that PopC is a periplasmic protein and is secreted from the periplasm across the OM to the extracellular milieu. Moreover, we found that the PMF has a role in PopC secretion to the extracellular milieu. It remains an open question how PopC is translocated across the IM to the periplasm. Generally, periplasmic proteins are synthesized with an N-terminal cleavable signal peptide and translocated across the IM by the Sec or Tat systems[2,3]. However, PopC does not contain a signal

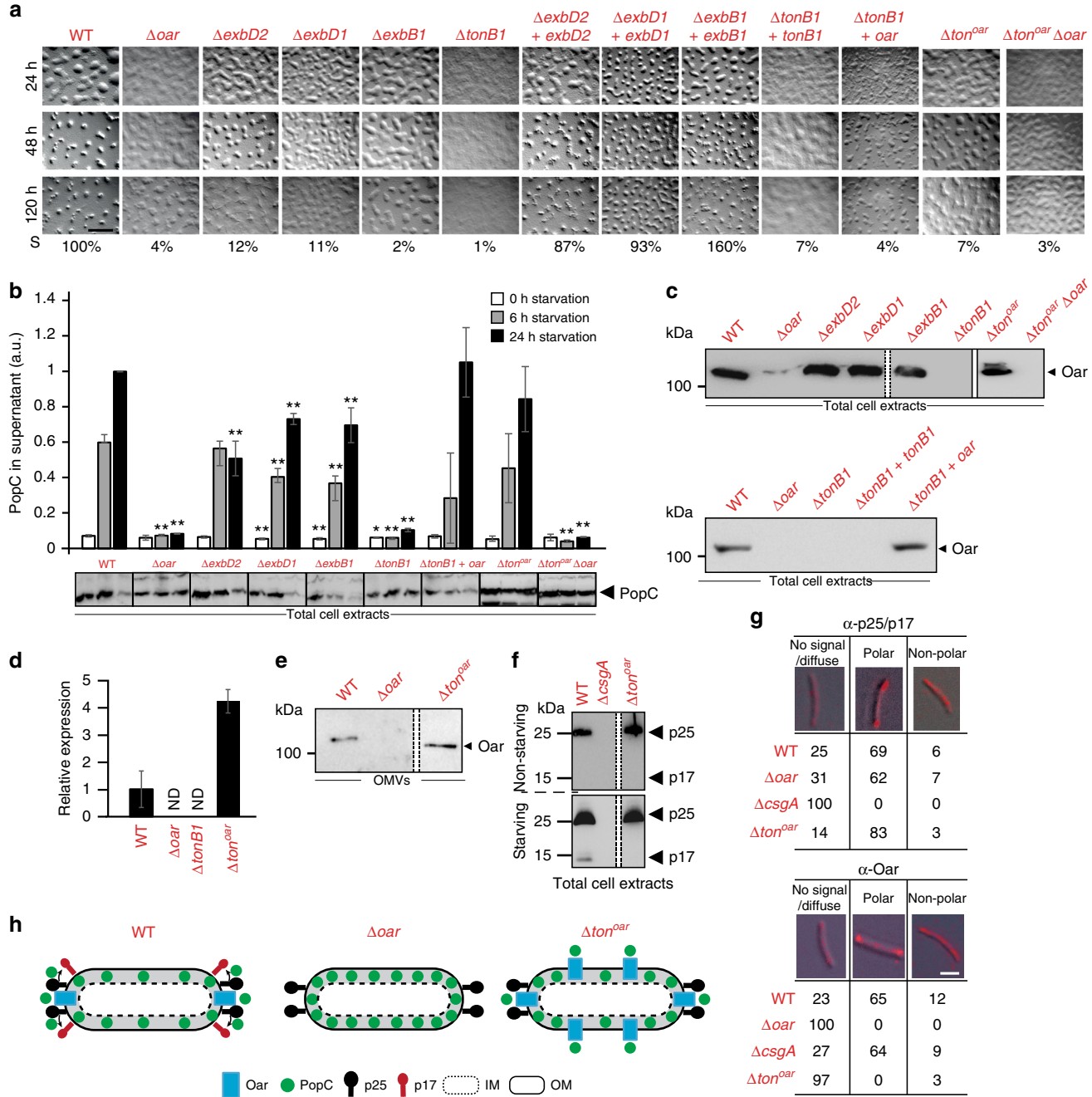

**Fig. 6** Characterization of Ton^Oar system mutants. **a** Developmental phenotype of strains of indicated genotypes. Strains were imaged at indicated time points of starvation. Sporulation (S) is expressed as percentage of WT. Scale bar, 1 mm. Images for WT and Δoar strains are the same as in Fig. 3c. **b** Accumulation of PopC in cell-free supernatants from strains of indicated genotypes. Experimental details and presentation as in Fig. 4a. n = 3–4. Error bars: s.d. **c** Immunoblot detection of Oar in cell extracts from non-starving cells of different Ton^Oar system mutants. White bar between dashed lines indicate lanes that have been removed from the same blot. White bar between solid lines indicate different blot. **d** oar expression in cell extracts from non-starving cells of the indicated strains. Total RNA was isolated and oar mRNA levels determined by qRT-PCR. Expression values are presented relative to the WT average. ND not detected. n = 2. Error bars: s.d. **e** Immunoblot detection of Oar in OMV in strains of the indicated genotypes under non-starving conditions. The blot for WT and Δoar mutant is the same as in Fig. 3g. White bar between dashed lines indicate lanes that have been removed. **f** p17 is not generated in Δton^oar mutant. Immunoblots of p25 and p17 accumulation in total cell extracts from non-starving and 6 h starving cells of indicated genotypes. Blot for WT and ΔcsgA is the same as in Fig. 4b. White bar between dashed lines indicate lanes that have been removed. **g** Polar localization of Oar depends on Ton^Oar system. Localization of p25/p17 and Oar were determined by immunofluorescence microscopy. Numbers indicate % of cells with that localization pattern. Representative cells shown. Non-polar pattern indicate presence of clusters along cell length. For the Δton^oar mutant a signal was mostly not detected with α-Oar antibodies. However, Oar accumulates at WT levels and in the OM in this strain (see **c**; upper and **e**). 100 < N < 184. Scale bar, 2 μm. **h** Schematic of model of Oar and p25 localization as well as p17 production in WT, Δoar, and Δton^oar strains with proposed polar PopC secretion. Source data for **b**–**f** are provided as a Source Data file

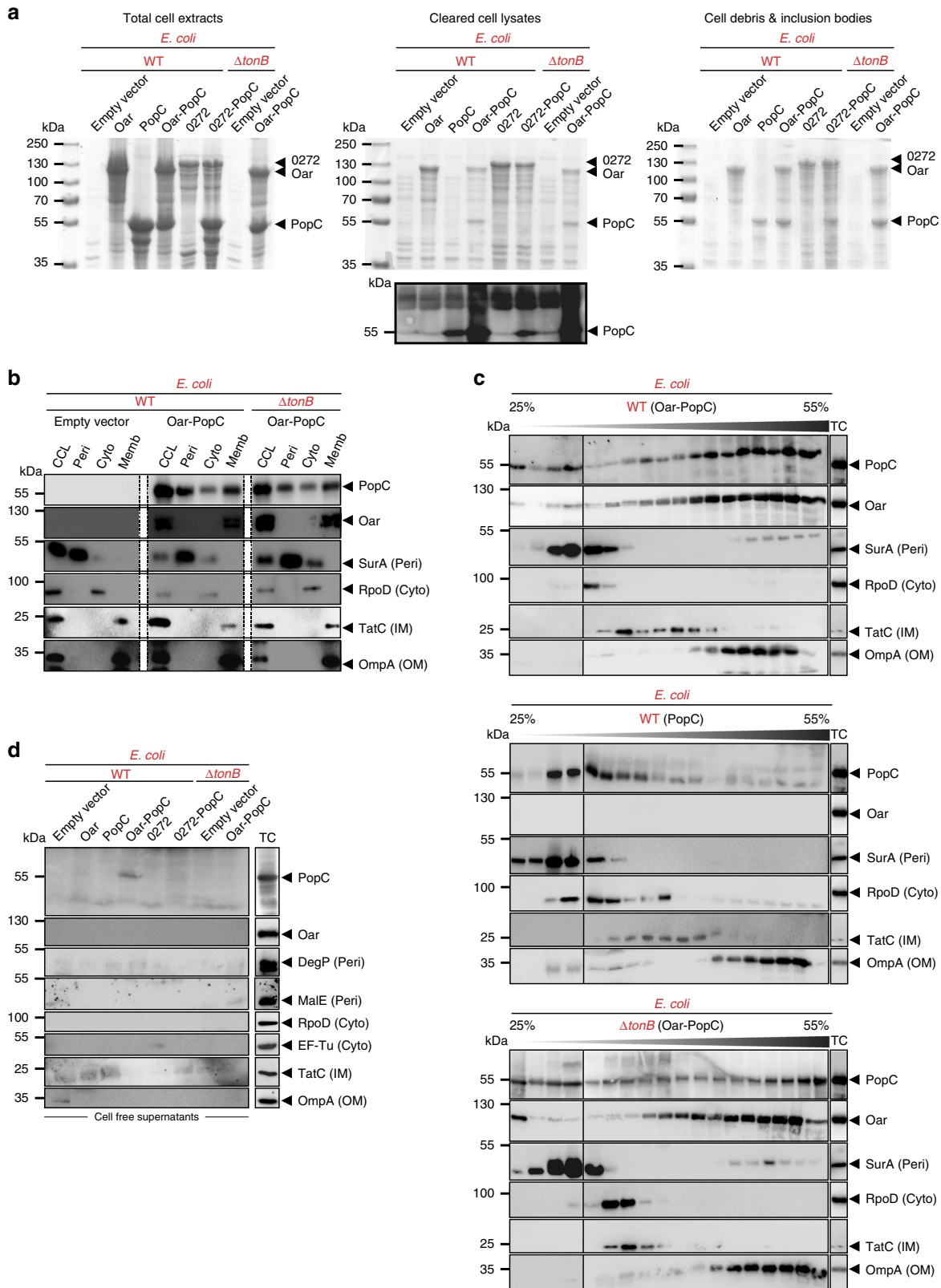

peptide and is secreted as a full-length protein by *M. xanthus*. Interestingly, PopC accumulates in the periplasm of *M. xanthus* and *E. coli* suggesting that the mechanism underlying translocation across IM is conserved in both *M. xanthus* and *E. coli*. Further work is required to dissect the mechanism by which PopC is translocated across the IM.

Second, we demonstrate that the TBDT Oar is required for secretion of PopC across the OM by *M. xanthus* as well as by *E. coli*. Moreover, heterologous expression experiments in *E. coli* mutants lacking components of the Ton system demonstrated that all three components of a functional Ton system are required for PopC secretion across the OM. Similar to TBDTs involved in

**Fig. 7** Oar- and TonB-dependent PopC secretion by *E. coli*. **a** PopC, Oar, and MXAN_0272 accumulation in *E. coli*. Coomassie brilliant blue-stained SDS-PAGE gels loaded with total cell extracts, cleared cell lysates, or cell debris/inclusion bodies after induction of the indicated proteins in the indicated *E. coli* strains. PopC immunoblot of cleared cell lysates shown below the gel. **b** Immunoblots of cell fractions from indicated *E. coli* strains co-expressing Oar and PopC. CCL, cleared cell lysate, Peri, periplasm, Cyto, cytoplasm, Memb, total membrane fraction. White bars: lanes removed from the same blot. WT strain with empty vector was used as control. **c** Immunoblots from sucrose gradient centrifugation separation of IM and OM from indicated *E. coli* strains expressing Oar and PopC together or PopC alone. Sucrose concentration indicated as % and gray triangle. Solid lines separate different blots. **d** Immunoblot detection of indicated proteins in cell-free supernatants of *E. coli* strains shown in **a**. PopC detected in the cell-free supernatant corresponds to ~0.2 % of PopC in cleared cell lysates. **b–d** SurA, DegP, MalE, RpoD, EF-Tu, TatC, and OmpA are markers for the indicated subcellular fractions. **c, d** Lane marked TC: total cell extracts from WT cells. Source data are provided as a Source Data file

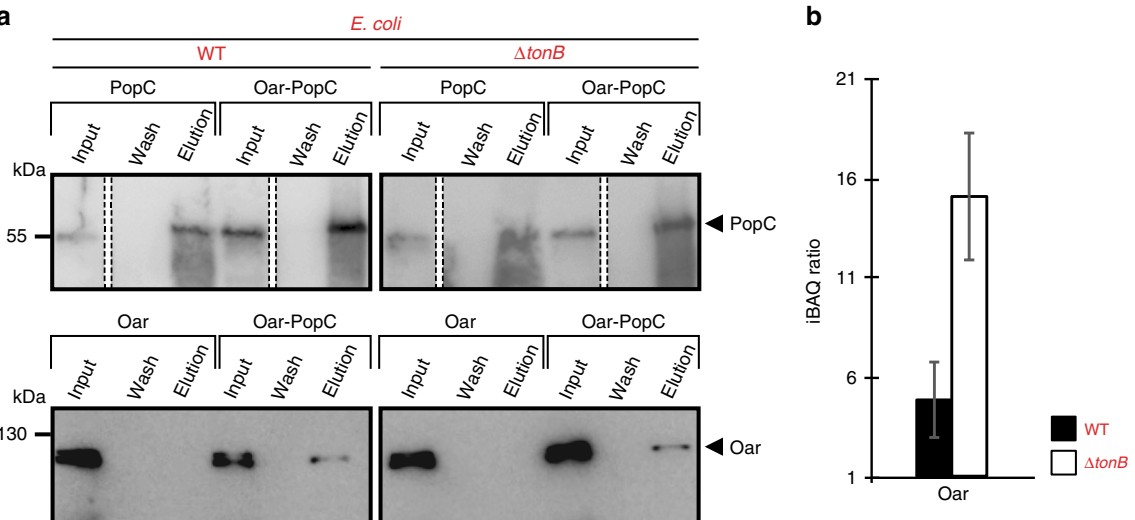

**Fig. 8** PopC and Oar interact in *E. coli*. **a** PopC and Oar interact. α-PopC antibodies were used in co-IP experiments on isolated membranes of *E. coli* strains of the indicated genotypes expressing PopC and/or Oar. Proteins were detected by immunoblotting for PopC (upper) and Oar (lower). White bars between dashed lines indicate lanes that have been removed. **b** Enrichment of Oar in co-IP experiments with α-PopC antibodies in strains co-expressing PopC and Oar. Samples from **a** were analyzed by LFQ-MS, iBAQ values calculated (Methods), and iBAQ ratios calculated as the ratio between Oar in the presence and absence of PopC. Source data are provided as a Source Data file

import processes across the OM, Oar contains an N-terminal plug domain with a TonB box. As expected based on comparison to TBDT importers, our data support that the TonB box in the plug domain is important for PopC secretion, supporting that secretion of PopC across the OM depends on the Ton system in the IM and the TBDT Oar in the OM.

Third, four sets of experimental evidence support that Oar and PopC interact directly. (i) Oar and PopC can be crosslinked resulting in the association of PopC with the membrane fraction in *M. xanthus*. (ii) PopC solubility increased when co-expressed with Oar in *E. coli*. (iii) PopC associated with the OM in *E. coli* when co-expressed with Oar. (iv) Oar was enriched in co-IP experiments using α-PopC antibodies in *E. coli* cells expressing Oar together with PopC compared to *E. coli* cells only expressing Oar. In the latter experiments, Oar was more highly enriched in the Δ*tonB* mutant than in WT cells (Fig. 8b). We speculate that in the Δ*tonB* mutant there is no translocation across the OM and, therefore, Oar-PopC intermediates are trapped and the Oar-PopC interaction becomes more evident.

TBDTs involved in import processes across the OM bind their ligands to their cell surface-exposed parts. Ligand binding leads to conformational changes in the plug domain of the TBDT that allow it to interact with the CTD of TonB, resulting in ligand import across the OM[18,37]. While the precise transport mechanism remains unclear, it has been proposed to involve rearrangements of the plug domain within the β-barrel to allow threading of the ligand through the lumen of the β-barrel[18,37]. For instance, the *Pseudomonas aeruginosa* TBDT FpvA1 was

recently shown to import the 74 kDa bacteriocin PyoS2 across OM by a mechanism that involves threading of PyoS2 through the lumen of the β-barrel while the plug remains partially inside the lumen[38]. In all structurally characterized TBDTs the lumen of the β-barrel has a diameter of 35–40 Å[18,39]. Oar is predicted to have a structure and lumen diameter similar to these TBDTs (Fig. 3b and Supplementary Table 1). Although our data do not allow us to rule out the possibility that PopC once bound to Oar would be "handed-over" to another secretion machinery, which would have to be conserved in the OM of both *M. xanthus* and *E. coli*, our data support a simpler scenario in which Oar together with a functional Ton system are required and sufficient for PopC secretion across the OM, and that Oar together with a functional Ton system may represent a novel system for protein secretion across the OM. It is important to emphasize that it has not been demonstrated that PopC passes through the Oar β-barrel. Nevertheless, combining our functional data with the mechanistic insights from TBDTs involved in import, it is tempting to speculate that secretion of the most likely unfolded 50.8 kDa PopC protein by Oar occurs similarly to the uptake of bacteriocins by TBDTs albeit in reverse. In *M. xanthus*, PopC and Oar accumulate in the periplasm and OM, respectively in non-starving cells. Moreover, preliminary gene expression analyses suggest that the proteins of the Ton$^{Oar}$ system accumulate in non-starving cells. In addition, cells treated with chloramphenicol and then exposed to starvation secrete PopC similarly to untreated cells[8] supporting that the system responsible for PopC secretion is present in non-starving cells. We are currently investigating how

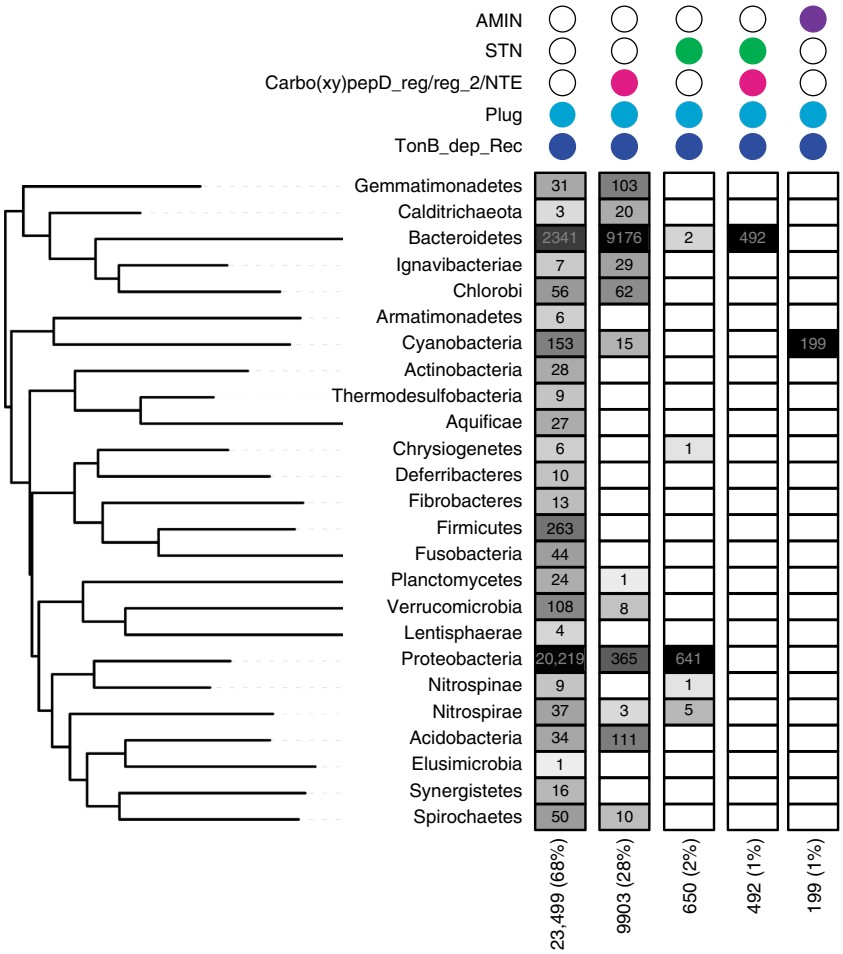

**Fig. 9** Distribution of TBDTs in different phyla according to N-terminal domain architecture. The five identified domain architectures are represented by colored circles according to the presence of the indicated Pfam domains. Numbers indicate proteins per taxon with the indicated architecture. The gray scale represents frequency of proteins (maximum black and minimum white). Pfam domains: TBDT β-barrel domain (PF00593; TonB_dep_Rec), TBDT plug domain (PF07715; Plug), Carboxypeptidase regulatory-like domain (PF13620; CarboxypepD_reg), CarboxypepD_reg-like domain (PF13715; CarbopepD_reg_2), Secretin and TonB N terminus short domain (PF07660; STN), and AMIN domain (PF11741; AMIN). The hidden Markov models of PF13620 and PF13715 largely overlap and, therefore, proteins with these domains were combined. Carbo(xy)pepD_reg/reg_2 is short for the two Pfam domains PF13620 and PF13715. The Oar domain referred to as NTE throughout the text is annotated as a PF13620 domain in Pfam. The phylogenetic tree was generated using representative species for each taxon (Supplementary Table 4)

the Oar-Ton[Oar] system is licensed to support secretion of PopC in response to starvation. Also, the mechanism underlying the slow PopC secretion kinetics warrants further analyses.

TBDTs are among the most widespread OM proteins in Gram-negative bacteria[40] (see also Fig. 9), with some species, e.g. abundant *Bacteroides* spp in the human gut microbiome[41], encoding more than 100 TBDTs[33] raising the question how widespread TBDTs involved in protein secretion could be. To begin to address this question, we performed bioinformatic analyses in which we identified >30,000 TBDTs of which the majority have unknown functions. These proteins can be divided into five groups based on their N-terminal domain architecture (Fig. 9). Because the WT Oar protein as well as an Oar variant lacking the NTE domain support PopC secretion (Fig. 5c), we conclude that the two Oar variants that encompass the two dominant N-terminal architectures (Fig. 9) both support PopC secretion. Interestingly, TBDT importers include proteins without additional N-terminal domains, e.g. the vitamin B12 importer BtuB in *E. coli*[18] as well as proteins with a domain architecture similar to Oar, e.g. the carbohydrate importer SusC from *Bacteroides thetaiotaomicron*[42]. Thus, the sequence features that may

distinguish a TBDT involved in secretion from one involved in import are not currently known and their identification awaits the characterization of additional TBDTs involved in protein secretion. However, based on the data presented here, we speculate that some of the many uncharacterized TBDTs may not only be involved in import but additionally, or exclusively, in protein secretion across the OM. Finally, the work presented here illustrates a remarkable functional analogy between TBDTs/Ton systems and ABC transporters[43], i.e. these two systems support export as well as import processes of varied substrates across the OM and IM, respectively, thus, underscoring the striking evolutionary malleability of membrane transport systems.

## Methods

**Bacterial strains, plasmids, growth media, and chemicals**. Strains, plasmids and oligonucleotides used in this study are listed in Supplementary Table 2 and 3. All *M. xanthus* strains are derivatives of DK101[44] except the two negative control strains used for motility experiments, which are derivatives of DK1622[45]. In-frame deletion mutants were generated as described[46]. Plasmids for ectopic expression of genes in *M. xanthus* were integrated by site-specific recombination at the *attB* site and the relevant genes expressed from the *pilA* promoter. Point mutations were generated using the Q5® Site-Directed Mutagenesis Kit (New England Biolabs)

following the manufacturer's recommendations. To increase detection of p17, all strains in Figs. 4b, 6f ectopically overexpressed *csgA* from the *pilA* promoter. All strains were confirmed by PCR.

*M. xanthus* was grown in liquid 1% CTT medium (1% casitone, 10 mM Tris-Cl (pH 7.6), 1 mM $K_2HPO_4/KH_2PO_4$ (pH 7.6), and 8 mM $MgSO_4$) with agitation or on 1% CTT 1.5% agar plates at 32 °C[44]. Kanamycin was used at concentrations of 80 µg ml$^{-1}$. Chloramphenicol was used at a concentration of 25 µg ml$^{-1}$. Cell growth was measured as an increase in optical density at 550 nm using an Ultrospec 2100 pro spectrophotometer (Amersham Biosciences, München). For motility assays, cells were grown in CTT medium to a density of $5 \times 10^8$ cells per ml, harvested, and resuspended in 1% CTT to a calculated density of $7 \times 10^9$ cells per ml. Five-microliter aliquots of cell suspensions were placed on 0.5% (to score type IV pili-dependent motility[47]) and 1.5% agar (to score gliding motility[47]) supplemented with 0.5% CTT (0.5% casitone, 10 mM Tris-Cl (pH 7.6), 1 mM $K_2HPO_4/KH_2PO_4$ (pH 7.6), and 8 mM $MgSO_4$) and incubated at 32 °C. After 24 h, colony edges were observed using a Leica MZ8 stereomicroscope or a Leica IMB/E inverted microscope and imaged by using Leica DFC280 or DFC350FX charge-coupled-device cameras, respectively. For development, cells were grown exponentially as described above, harvested, and resuspended in MC7 buffer (10 mM MOPS (pH 7.0) and 1 mM $CaCl_2$) to a calculated density of $7 \times 10^9$ cells per ml. Twenty-microliter aliquots of cells were placed on 1.5% agar TPM (10 mM Tris-HCl (pH 7.6), 1 mM $K_2HPO_4/KH_2PO_4$ (pH 7.6), and 8 mM $MgSO_4$). Cells were visualized at the indicated time points using a Leica MZ8 stereomicroscope and imaged using a Leica DFC280 camera. Sporulation levels were determined after development for 120 h on TPM agar as the number of sonication- and heat-resistant (55 °C for 2 h) spores relative to WT. To determine PopC secreted to the cell-free supernatant, cells were starved in suspension by first harvesting them from exponentially grown cultures with a maximum density of $5 \times 10^8$ cells per ml followed by resuspension in MC7 buffer to a calculated density of $10^9$ cells per ml. Starvation was performed at 32 °C with agitation in the presence of protease inhibitors (Complete Mini from Roche: henceforth PI) as described[7,8], and samples collected at the indicated time points.

*E. coli* strains were grown in LB broth in the presence of relevant antibiotics[48]. All plasmids were propagated in *E. coli* Mach1. Proteins were expressed in Rosetta-2(DE3) by adding 1 mM of isopropyl β-D-1-thiogalactopyranoside (IPTG) to exponentially growing cultures at a density of $3.5 \times 10^8$ cells per ml, and BACTH assays were conducted in BTH101. Mutants from the Keio collection[49] (Supplementary Table 2) were used as negative controls for identification of relevant proteins by immunoblotting. ΔtonB, ΔexbB, and ΔexbD mutations in Rosetta-2(DE3) were generated by P1 transduction, using as donors the corresponding strains from the Keio collection followed by removal of the kanamycin resistance cassette using the pCP20 plasmid encoding the yeast FPL recombinase[50].

**Cell fractionation.** Subcellular fractionation of *M. xanthus* cells was done using cells grown exponentially in liquid CTT or using cells that had been starved in suspension for 6 h in MC7 in the presence of PI. To isolate cell-free supernatants, cells were harvested at room temperature (RT) (15,000 × *g* for 10 min). Supernatants were filtered through a 0.22 µm sterile filter (Millipore, Schwalbach, Germany) and centrifuged (150,000 × *g* for 2 h at 4 °C) to pellet OMV. The resulting cell-free supernatants were then precipitated with 20% of ice-cold trichloroacetic acid (TCA) for 30 min, followed by 15 min centrifugation (15,000 × *g* at 4 °C) and two wash steps with ice-cold acetone with centrifugation steps between washes (5 min at 4 °C, max. speed in a tabletop centrifuge when using 1.5 ml tubes, or 10 min at 4 °C, 15,000 × *g* when using 15–50 ml tubes). Pellets were air-dried for 15–30 min and resuspended in SDS-loading buffer.

To separate periplasmic, cytoplasmic, and membrane proteins, pelleted cells obtained from the centrifugation to isolate cell-free supernatants were washed in 1/3 of the original volume with saline phosphate buffer pH 7.4[48] (PBS: 8 mM $Na_2HPO_4$, 2 mM $KH_2PO_4$, 137 mM NaCl, and 2.7 mM KCl, pH 7.4.), pelleted at RT (15,000 × *g* for 10 min) and resuspended very gently with a glass pipette to $3–4 \times 10^{10}$ cells per ml. An aliquot was kept as total cell fraction, 2 ml were used to separate periplasmic from cytoplasmic proteins and 3 ml to separate membrane proteins. To separate periplasmic and cytoplasmic proteins, cells were pelleted (8000 × *g* for 10 min at RT) and washed twice with 25 mM Tris-HCl (pH 7.6) buffer (pelleting cells between washes by centrifugation at 8000 × *g* for 10 min at RT). Cells were resuspended in 1 ml 12.5 mM Tris-HCl (pH 7.6) buffer with 1 mM EDTA, 20% sucrose, 1 mg ml$^{-1}$ of lysozyme (Sigma-Aldrich), and PI, incubated for 10 min at RT and pelleted at 8000 × *g* for 10 min at RT. Supernatants were discarded and cells resuspended in ice-cold 0.5 mM $MgSO_4$ with PI, incubated 10 min on ice, and pelleted at 10,000 × *g* for 10 min at 4 °C. Pellets were kept to isolate cytoplasmic proteins and the supernatants were filtered through a 0.22 µm sterile filter, ultracentrifuged at 40,000 × *g* for 1 h at 4 °C to pellet membranes, and finally, supernatants containing periplasmic enriched fractions were precipitated with TCA as described above. The cell pellets kept to isolate cytoplasmic proteins were washed once with 25 mM Tris-HCl (pH 7.6) buffer + PI, sonicated, centrifuged at 10,000 *g* for 15 min at 4 °C and the supernatants filtered through a 0.22 µm sterile filter. Then the filtered supernatants were ultracentrifuged at 40,000 × *g* for 1 h at 4 °C to pellet membranes, and the resultant supernatants enriched in cytoplasmic proteins were precipitated with TCA as described above. The 3 ml of cells stored to

separate membrane proteins were pelleted (8000 × *g* for 10 min at 4 °C), resuspended in 3 ml of 25 mM Tris-HCl (pH 7.6) buffer supplemented with DNAse I and RNAse A (0.1 mg ml$^{-1}$, Sigma-Aldrich) + PI, sonicated and centrifuged at 5000 × *g* for 20 min at 4 °C. Supernatants were collected and ultracentrifuged at 40,000 × *g* for 1 h at 4 °C to pellet membranes (fraction enriched in membrane proteins). Pellet from fractions enriched for cell-free supernatant proteins, periplasmic, cytoplasmic, and membrane proteins were resuspended in SDS-loading buffer. In Fig. 1a, b, total cell extracts, cell-free supernatant, periplasm, cytoplasm, and membranes from $1–2 \times 10^6$ cells, $10^{10}$ cells, $10^8$ cells, $10^8$ cells, and $10^8$ cells, respectively, were loaded. In Supplementary Fig. 1c, cell-free supernatant, periplasm, and cytoplasm were loaded from $10^{10}$, $10^7$, and $10^8$ cells, respectively. All fractions were analyzed by immunoblotting as described[48] using rabbit polyclonal antibodies against PopC[7] (dilution. (dil.) 1:1000), Oar[51] (dil. 1:10000), GltC[51] (dil. 1:5000), PilB[52] (dil. 1:5000), PilC[53] (dil. 1:5000), and GltA[51] (dil. 1:5000), and as secondary antibodies goat anti-rabbit immunoglobulin G peroxidase conjugate (Sigma-Aldrich, cat.nr. A8275) (dil. 1:1000).

Separation of cleared cell lysates and cell debris/inclusion bodies from total cell extracts of *E.coli* was done following sonication of cells resuspended in 0.2 M Tris-HCl (pH 7.6) by centrifugation at 5000 × *g* for 15 min at 4 °C. The supernatant after this centrifugation contains cleared cell lysates and the pellet contains cell debris/inclusion bodies. Fractionation of *E. coli* was done by separating periplasmic, cytoplasmic, and membrane proteins as described[54] with minor modifications: cells were pelleted and resuspended in 0.2 M Tris-HCl (pH 7.6), 20% sucrose, 1 mM EDTA, 1 mg ml$^{-1}$ lysozyme, and PI, incubated for 5 min at RT, and pelleted. Cells were resuspended in water + PI and incubated on ice, centrifuged at 100,000 × *g* for 1 h at 4 °C and the supernatant was stored as the periplasmic fraction. The pellet was resuspended in ice-cold 10 mM Tris-HCl (pH 7.6), 5 mM EDTA, and 0.2 mM DTT supplemented with 1 mg ml$^{-1}$ DNAse I. Cells were sonicated and unbroken cells were spun down by centrifugation at 5000 × *g* for 20 min at 4 °C. The supernatant was centrifuged at 100,000 × *g* for 2 h at 4 °C. Supernatant after this centrifugation contains the cytoplasmic fraction and the pellet contains the crude membranes. Sucrose gradient centrifugation was done according to Holkenbrink et al.[55] with TCA protein precipitation following the protocol described above. Cell-free supernatants were collected and processed as described above for *M. xanthus*. Fractions were analyzed by immunoblotting using rabbit polyclonal antibodies against SurA (Acris Antibodies GmbH, cat.nr. AS10798) (dil. 1:5000), MalE[56] (1:30,000), and DegP (1:30,000) (gift of T. Silhavy) for periplasm, RpoD (Santa Cruz Biotech, cat.nr SC56768) (dil. 1:10000) and EF-Tu (HycultBiotech, cat. nr HM6010) (1:2000) for cytoplasm, TatC[57] (1:10000) for IM (gift of T. Palmer), and OmpA[58] (1:30000) for OM (gift of T. Silhavy). In Fig. 7b, total cell extracts from $10^8$ cells, and periplasm, cytoplasm, and membranes from $10^9$ cells were loaded. In Fig. 7d, cell-free supernatants from $5 \times 10^{10}$ cells were loaded.

**Protein detection.** Secreted PopC was detected using an enzyme-linked immunosorbent assay (ELISA) procedure with α-PopC antibodies as described[7,8]. Differences in PopC secretion detected by ELISA were determined using a two sample *t*-test (two-tailed, unequal variances), and considering levels of significance ≤ 0.05. Sample size (N) is indicated in each figure (with 2–3 technical replicates each). In total cell extracts, PopC, Oar, and p17/p25 were detected by immunoblotting using rabbit polyclonal α-PopC[7] (dil. 1:1000), α-Oar[51] (1:10,000), and α-p17/p25 (dil. 1:1000) antibodies, which recognize the C terminus of p17 and p25[12].

**Quantification of PopC secretion in the presence of drugs.** Quantification of PopC in cell-free supernatants after addition of CCCP, valinomycin, nigericin or arsenate, was done using ELISA as described[8], with a wash and recovery step added. Drugs were added at the indicated concentrations after cells were resuspended in MC7 buffer to a calculated density of $10^9$ cells per ml and time point 0 h was collected. CCCP, valinomycin, and nigericin were dissolved in dimethyl sulfoxide (DMSO), therefore, an MC7 + DMSO control sample was used for comparison. After collecting the last sample in the presence of drug, cells were pelleted (15,000 × *g* for 10 min at RT), washed twice with MC7 buffer, and resuspended in an equal volume of fresh MC7 without any drug or chemical added. Differences in PopC secreted were determined using a two sample *t*-test (two-tailed, unequal variances), and considering levels of significance ≤ 0.05. Sample size (N) is indicated in Fig. 2 (with three technical replicates each).

**Measurement of cellular ATP content.** ATP content was measured following the instructions provided by the manufacturer for the ATP Bioluminescence Assay Kit CLS II (Sigma-Aldrich), and using black polystyrol LumiNunc 96-well flat bottom plates (Thermo Fisher Scientific) and an Infinite 200 PRO plate reader (Tecan) with injector module, to trigger fast kinetic reactions in luminescence mode. Differences in ATP content were determined by using a two sample *t*-test (two-tailed, unequal variances), and considering levels of significance ≤ 0.05. N = 4 (with 3 technical replicates each).

**Fluorescence microscopy.** Immunofluorescence microscopy was performed essentially as described[53]. Briefly, *M. xanthus* cells were resuspended to a calculated density of $7.0 \times 10^9$ cells per ml in PBS from TPM agar plates after 6 h of starvation. Cells were fixed with 1.6% paraformaldehyde and 0.008% glutaraldehyde for 20

min on freshly prepared poly L-lysine-treated 12-well diagnostic slides (Thermo Fischer Scientific). Cells were permeabilized with GTE buffer (50 mM glucose, 20 mM Tris, 10 mM EDTA, pH 7.5) for 10 min, washed and blocked for 30 min at RT. Cells were probed with relevant rabbit polyclonal antibodies at RT for 90 min (α-Oar, dil. 1:2000) and 30 min (α-p17/p25, dil. 1:200). Alexa Fluor 594 goat anti-rabbit IgG (Thermo Fischer Scientific, cat.nr LSA11037) (dil. 1:200) was added as secondary antibody for 1 h after washing away the primary antibody. Secondary antibody was washed away followed by addition of Slow Fade Anti Fade Reagent (Molecular Probes) to each well. For each strain, at least 100 cells were analyzed. Cells were observed using a Leica DMI6000B microscope with a Hamamatsu Flash 4.0 camera. Images were recorded with Leica MM AF software package and processed with Metamorph (Molecular Devices).

**Crosslinking.** For in vivo crosslinking experiments, *M. xanthus* cells were starved for 6 h in MC7 buffer in suspension in the presence of PI at a calculated density of $10^9$ cells per ml. Cells were pelleted at $15,000 \times g$ for 10 min at RT and washed in PBS. After a second centrifugation step under the same conditions, cells were resuspended in the same volume of PBS. Where indicated, DMSO or the cross-linking agent DSP (Thermo Fischer Scientific) resuspended in DMSO to 0.25 mM final concentration were added and cells were incubated for 30 min at 30 °C with gentle agitation. The crosslinking reaction was quenched with 50 mM Tris-HCl (pH 7.6) for 15 min at RT, and cells were pelleted at $15,000 \times g$ for 10 min at RT. Cell pellets were resuspended in 1/10 of the volume of PBS, sonicated, and centrifuged at $5000 \, g$ for 10 min at 4 °C to remove cell debris. Supernatants were filtered through a 0.22 μm sterile filter and ultracentrifuged at $40,000 \times g$ for 1 h at 4 °C. Pellets containing membrane fraction (washed once more in PBS) were resuspended in loading buffer (62.5 mM Tris-HCl (pH 7.6), 4% SDS, 20% glycerol, and 0.25% bromophenol blue) with or without 50 mM DTT and separated in Any kD™ Mini-PROTEAN® TGX™ Precast Protein Gels (Bio-Rad). Proteins were detected by immunoblotting using α-PopC and α-Oar antibodies[7,51].

**Bacterial two-hybrid experiments.** BACTH experiments were performed as described[59]. Fragments containing the periplasmic N-terminal region of Oar that bears NTE and plug domains (Oar$_{24-248}$) and CTD of TonB1 (TonB1$_{147-250}$), MXAN_0276 (MXAN_0276$_{167-269}$), and MXAN_0820 (MXAN_0820$_{163-266}$) from *M. xanthus* or *E. coli* (TonB$_{139-239}$) were cloned into the appropriate vectors to construct N-terminal and C-terminal fusions with the 25-kDa N-terminal or the 18-kDa C-terminal adenylate cyclase (CyaA) fragments. cAMP production by reconstituted CyaA was qualitatively assessed by the formation of blue color on LB agar supplemented with 40 μg ml$^{-1}$ 5-bromo-4-chloro-3-indolyl-β-D-galactopyranoside (X-Gal) and 0.5 mM IPTG or quantitatively by measuring β-galactosidase activity as follows. Five microliters of LB media supplemented with kanamycin, ampicillin, and 0.5 mM IPTG were inoculated with one colony and grown overnight at 30 °C. Cells were sonicated and 2 ml of cell extract was centrifuged at $15,000 \times g$ for 5 min at 4 °C. Supernatants containing β-galactosidase were collected and used for enzyme activity quantification after measuring protein content by the Microassay Procedure of the Bio-Rad protein assay (Bio-Rad). Enzymatic activity was measured in Z-buffer, pH 7.0 (60 mM Na$_2$HPO$_4$·7H$_2$O, 40 mM NaH$_2$PO$_4$·H$_2$O, 10 mM KCl, 1 mM MgSO$_4$, and 50 mM β-mercaptoethanol) supplemented with 1 mg ml$^{-1}$ of freshly made ortho-nitrophenyl-β-galactoside (ONPG), as increased absorbance at 420 nm. Reactions were stopped by adding 1 M Na$_2$CO$_3$ and activity was expressed as specific enzymatic activity (nmol ONPG hydrolyzed per mg protein per min).

**Measurement of ppGpp in vivo.** ppGpp detection was performed as previously described[60] with some modifications. To label cells, they were grown in liquid 0.5% CTT in the presence of radioactive orthophosphoric acid ($^{32}$PO$_4$) at 100 μCi ml$^{-1}$ of culture. Cells were starved in TPM buffer supplemented with 100 μCi ml$^{-1}$ of $^{32}$PO$_4$ and 0.5 ml of starving culture was collected at indicated time points. For nucleotide extraction, samples were placed on ice and heated for at least 15 min with 1 M formic acid. Extracts were stored at −20 °C. Acid extracts were thawed if frozen and centrifuged at $12,000 \times g$ for 5 min. Thirty microliters of supernatant were spotted on polyethylenimine-cellulose plates (Sigma) and samples were run with 1.5 M KH$_2$PO$_4$ (pH 3.5) as the mobile phase, and developed until the solvent front reach the end of the sheet. Plates were dried prior to exposing a phosphor-imaging screen (Molecular Dynamics). Data were collected using a STORM 840 scanner (Amersham Biosciences) and analyzed using ImageJ 1.46r.

**RNA isolation and qRT-PCR.** Total RNA was isolated using TRIzol™ Reagent (Thermo Fischer Scientific) following the instructions provided by the manufacturer. RNA was treated with DNase I (Thermo Fischer Scientific) and purified by precipitation (0.3 M sodium acetate (pH 5.2) and 2.5 volumes 99% ethanol) followed by centrifugation at max. speed for 30 min at 4 °C. RNA was confirmed to be DNA-free by PCR analysis. One microgram of DNA-free total RNA was used to synthesize cDNA with the High capacity cDNA Archive kit (Applied Biosystems) using random hexamers as primers. qRT-PCR was performed in 25 μl reaction volume using SYBR green PCR master mix (Applied Biosystems) and 0.1 μM primers specific to the *oar* gene in a 7500 Real Time PCR System (Applied Biosystems). Each reaction was performed in triplicate with two biological replicates

per strain. Gene-specific primers were designed using Primer Express 2.0.0 and tested prior to qRT-PCR for amplification efficiency on 10-fold serial dilutions of genomic DNA. Relative gene expression levels were calculated using the comparative Ct method.

**Co-IP and LFQ-MS.** Membranes from *E. coli* were obtained as described in the section "Cell fractionation" (crude membranes). Each isolated membrane fraction was resuspended in 2 ml buffer containing 50 mH HEPES (pH 7.5), 150 mM NaCl, 5 mM EDTA, 1% sodium lauroylsarcosinate, and PI.

Co-IP experiments were done essentially as described[61]. Briefly, 25 μl undiluted α-PopC antibodies were added to 1.8 ml of isolated membranes, and incubated for 1 h at 4 °C on a rotating shaker. Then 20 μl of magnetic protein G beads (Thermo Fischer Scientific) were added for 1 h to capture antibodies. Beads were then washed four times with 700 μl of 100 mM ammoniumbicarbonate (Sigma-Aldrich). For elution, either elution buffer (1.0 M urea and 100 mM ammoniumbicarbonate) was added to the beads, or an on-bead digestion was performed by adding 100 μl trypsin-containing elution buffer 1 (1.0 M urea, 100 mM ammoniumbicarbonate, and 2 μg trypsin (Promega)) to each sample. After 30 min incubation at 30 °C, the supernatant containing (un)digested proteins was collected. In Fig. 8a, input material from $5 \times 10^8$ cells, and wash and elution fractions from $5 \times 10^{11}$ cells were loaded. For liquid chromatography-mass spectrometry (LC-MS) analysis, beads were washed twice with elution buffer 2 (1.0 M urea, 100 mM ammoniumbicarbonate, 5 mM Tris(2-carboxyethyl)phosphin (TCEP)) (Thermo Fischer Scientific) and added to the first elution fraction. Digestion was allowed to proceed overnight at 30 °C. Following digestion, the peptides were incubated with 10 mM iodoacetamide for 30 min at 25 °C in the dark. The peptides were acidified with trifluoroacetic acid (TFA, Thermo Fischer Scientific) and desalted using solid-phase extraction (SPE) on C18-Microspin columns (Harvard Apparatus). SPE columns were prepared by adding acetonitrile (ACN), followed by column equilibration with 0.1% TFA. Peptides were loaded on equilibrated Microspin columns and washed twice with 5% ACN/0.1% TFA. After peptide elution using 50% ACN/0.1% TFA, peptides were dried in a rotating concentrator (Thermo Fischer Scientific), reconstituted in 0.1% TFA and subjected to LC-MS analysis.

LC-MS analysis of the peptide samples was carried out on a Q-Exactive Plus instrument connected to an Ultimate 3000 RSLC nano and a nanospray flex ion source (all Thermo Fischer Scientific). Peptide separation was performed on a reverse-phase high-performance liquid chromatography column (75 μm × 42 cm) packed in-house with C18 resin (2.4μm, Dr. Maisch). The peptides were loaded onto a PepMap 100 precolumn (Thermo Fischer Scientific) and eluted by a linear ACN gradient from 2–35% solvent B over 60 min (solvent A: 0.15% formic acid; solvent B: 99.85% ACN in 0.15% formic acid). The flow rate was set to 300 nl min$^{-1}$. The peptides were analyzed in positive ion mode. The spray voltage was set to 2.5 kV, and the temperature of the heated capillary was set to 300 °C. Survey full-scan MS spectra ($m/z = 375–1500$) were acquired in the Orbitrap with a resolution of 70,000 full width at half maximum at a theoretical $m/z$ 200 after accumulation a maximum of $3 \times 10^6$ ions in the Orbitrap. Based on the survey scan up to 10 most intense ions were subjected to fragmentation using high collision dissociation at 27% normalized collision energy. Fragment spectra were acquired at 17,500 resolution. The ion accumulation time was set to 50 ms for both MS survey and tandem MS (MS/MS) scans. To increase the efficiency of MS/MS attempts, the charged state screening modus was enabled to exclude unassigned and singly charged ions. The dynamic exclusion duration was set to 30 s.

Label-free quantification (LFQ) of the samples was performed using MaxQuant (Version 1.5.3.17)[62]. For Andromeda database searches implemented in the MaxQuant environment, the protein databases for *M. xanthus* was downloaded from Uniprot[63] and searches were performed using the protein database. The search criteria were set as follows: full tryptic specificity was required (cleavage after lysine or arginine residues); two missed cleavages were allowed; carbamidomethylation (C) was set as fixed modification; oxidation (M) and deamidation (N, Q) as variable modification. MaxQuant was operated with default settings with the "Match-between-run" option.

To calculate protein enrichment in co-IP experiments, intensity-based absolute quantification (iBAQ) values[64] were calculated with MaxQuant. MaxQuant calculates protein intensities as a sum of all peptide intensities for a given protein. To obtain iBAQ values the protein intensity sum was divided by the number of theoretically observable peptides[64]. Calculated iBAQ values were normalized to protein input level (experimental procedure, see below) detected in the membrane fraction. Subsequently, iBAQ values were rescaled in order to compare different biological replicates.

To measure the input protein levels of PopC and Oar in the isolated membrane fractions, 200 μl of the sample was subjected to *Single-Pot* Solid-Phase-enhanced Sample Preparation (SP3)[65] as follows: in order to remove non-protein contaminants from the samples, a previously established strategy was modified for membrane fraction samples. A SP3 bead stock solution was prepared by mixing equal volumes of Sera-Mag magnetic carboxylate-modified particles and Speed-Bead Sera-Mag magnetic carboxylate-modified particles (both GE Healthcare). To the 200 μl of isolated membrane fraction, 5 mM TCEP was added and incubated for 15 min at 60 °C to reduce protein disulfide bonds, followed by alkylation of thiol groups using 10 mM iodoacetamide (Sigma-Aldrich) and incubated for 30 min at 25 °C in the dark. The reduced and alkylated sample was mixed with 250 μl ACN

and 4 µl of the prepared SP3 bead stock. After 15 min incubation, beads were washed twice with 70% ethanol and once with pure ACN. Beads were reconstituted in 10% ACN/50 mM ammoniumbicarbonate containing 1 µg sequencing grade modified trypsin (Promega) per sample. Tryptic digestion was carried out overnight at 30 °C. Following proteolytic digestion, the beads were separated on a magnet and the peptide containing supernatant was harvested. To increase the peptide recovery the beads were sonicated in the presence of 2% DMSO (Sigma-Aldrich). Finally, the peptides were purified by SPE (described above) and reconstituted in 0.1% TFA prior to LC-MS analysis (described above).

**Mapping of PopC N terminus**. Precipitated cell-free supernatants and cell pellets, respectively, were resuspended in 2% sodium lauroylsarcosinate in the presence of 5 mM TCEP and incubated for 15 min at 90 °C. Then iodoacetamide was added to 10 mM and samples incubated at 25 °C in the dark for 30 min. Fifty micrograms protein were transferred to a 30 kDa Microcon spin filter (Millipore), mixed with 8 M urea and centrifuged for 20 min at $13,000 \times g$. Samples were washed with 8 M urea and centrifuged twice as described to remove detergent and lipid-like material. After two washing steps, trypsin digestion was carried out in 1 M urea at 30 °C overnight. Upon digest completion, samples were acidified by adding 1% TFA and further desalted using C18-Microspin columns (Harvard Apparatus) as described.

LC-MS analysis was carried out as described in the co-IP section with the exception of an adjusted LC gradient. Peptides were separated by a linear ACN gradient from 2–32% solvent B over 105 min and to 50% B for another 35 min (solvent A: 0.15% formic acid; solvent B: 99.85% ACN in 0.15% formic acid). Peptide identification was performed with Proteome Discoverer (v. 1.4, Thermo Fischer Scientific) using SEQUEST HT (Thermo Fischer Scientific) and MASCOT (v 2.5, Matrix Science) as search engines against the Uniprot protein database of *M. xanthus*. The search criteria were set as follows: full tryptic specificity was required (cleavage after lysine or arginine residues); two missed cleavages were allowed; carbamidomethylation (C) was set as fixed modification; oxidation (M) and deamidation (N, Q), and acetylated protein N terminus as variable modification. The mass tolerance was set to 10 ppm and 0.02 Da for peptide precursors and fragment ions, respectively. For evaluation of the search results, the data were loaded into Scaffold 4 (Proteome Software) and a spectrum decoy-false discovery rate of 1% was adjusted. PopC coverage was directly exported from Scaffold.

**Bioinformatics**. BlastP[66] and Pfam[36] were used to identify protein domains. For similarity searches, we used BlastP and considered hits with *e*-values of 0.0001 or lower to be significant. SignalP v.4.1[67], TatP v.1.0[68], LipoP v.1.0[69], and Pred-Tat[70] were used to identify signal peptides with default gathering thresholds. TMHMM Server v.2.0[71] was used to predict transmembrane helices with default gathering thresholds. BOCTOPUS2[72] was used to predict β-strands and determine the topology of TBDTs, and TOPO2[73] was used to create the two-dimensional topology images. Phyre2 was used to compare Oar to TBDTs with available crystal structures[74]. T-coffee[75] was used to align plug domains and BioEdit v.7.0.5.3[76] to graphically represent those alignments.

To determine the distribution of TBDTs among different taxa, we extracted all protein identifiers from the Pfam database release 31.0 (http://pfam.xfam.org)[36] for those proteins that contain both the TBDT β-barrel domain (PF00593; TonB_dep_Rec) and the TonB plug domain (PF07715; Plug). In all, 34,893 bacterial proteins with five N-terminal domain architectures were identified that fulfill these criteria. The proteins were divided based on their N-terminal domain architecture, which is combinations of the following Pfam domains: carboxypeptidase regulatory-like domain (PF13620; CarboxypepD_reg), CarboxypepD_reg-like domain (PF13715; CarbopepD_reg_2), Secretin and TonB N terminus short domain (PF07660; STN), and AMIN domain (PF11741; AMIN). The hidden Markov models of domains PF13620 and PF13715 largely overlap and, therefore, these no distinction was made between these domains. All Pfam hits were taxonomically assigned to the phylum level at the NCBI taxonomy browser (https://www.ncbi.nlm.nih.gov/Taxonomy/)[77] and number of hits per taxon calculated. One hundred-fifty proteins could not be assigned to a phylum. Representative species for each taxon of interest (Supplementary Table 4) were used to obtain a 16S rRNA sequence from the RDP database, release 11, update 5 (http://rdp.cme.msu.edu)[78]. The phylogenetic tree was calculated with the phylogeny.fr pipeline, using default parameters (http://www.phylogeny.fr)[79]. For better visualization of the trees, Newick files were imported into the Interactive Tree Of Life platform (https://itol.embl.de)[80].

**Reporting summary**. Further information on experimental design is available in the Nature Research Reporting Summary linked to this article.

## Data availability

All data generated or analyzed during this study are included in this published article and its Supplementary Information files, or are available from the corresponding author upon request. The source data underlying Fig. 1a, b, 2, 3f, g, 4, 5c, d, 6b–f, 7, and 8 and Supplementary Figs. 1a, c, 2, 3b, 5 and 7b, e are provided as a Source Data file.

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

## Acknowledgements

We thank A. Konovalova, T. Palmer, T. J. Silhavy, and V. Sourjik for antibodies, strains, or plasmids. We are grateful to E. Bremer, A. Diepold, A. Filloux, and A. Konovalova for valuable discussions. This work was supported by the Max Planck Society and the Alexander von Humboldt Foundation.

## Author contributions

N.G.-S. designed and conceived the study and performed most of the experiments. T.G. performed immunoprecipitation experiments and mass spectrometry-based analyses. R.K. performed bioinformatic analyses to elucidate the distribution of TBDTs among different taxa. M.A.S.-P. assisted with *E. coli* fractionation and BACTH experiments. L.S.-A. supervised research and provided funding. N.G.-S and L.S.-A. analyzed and interpreted data and wrote the manuscript.

## Additional information

**Competing interests:** The authors declare no competing interests.

