## [Peer Review File · Nature Communications]

Reviewers' Comments:

Reviewer #1:

Remarks to the Author:

The manuscript by Nuria-Gomez et al describes the discovery of a TonB-dependent transporter (TBDT Oar) that functions as a protein secretion channel for the protease PopC. This is surprising and quite exciting because so far, TBDTs are known to be exclusively involved in OM uptake processes. The evidence presented for this is quite (almost too) elaborate and overall very convincing. If anything, the relevance of this important finding could be presented a bit more strongly, for example in the abstract, where the discovery of a novel secretion system is mentioned almost as an afterthought. As is, the study will form the basis for more detailed investigations into the mechanism of secretion by Oar. Like I said, this is a very nice paper and I only have some minor suggestions for clarification and improvement.

1. Is it possible to say how widespread the Oar TBDT secretion system is based on sequence conservation and perhaps phylogenetic analysis? Is it unique to *Myxococcus* spp? Does the Oar sequence have features that set it apart from regular "import" TBDTs?
2. The authors should clarify with wt strain of *M. xanthus* they are using. Although it is stated in Material and method, adding this information to the main text would help the reader.
3. Line 93: the authors state "because PopC is a soluble protein, lacks a signal peptide and is secreted as a full-length". They cite their own work (reference number 7). In that study, SignalP and TatP programs are used for an in silico search for a potential signal sequence. Authors claim that because no signal sequence is found by this software there is no signal sequence. This statement is also based in SDS and Western blots of PopC. However, signal sequences can be short, and a ~2 kDa difference might be difficult to see on a SDS gel. If the first ~20 amino acids are removed, is PopC still moved to the periplasm in *E. coli*? This would be simple to test. As it is, the statement that PopC secretion is Sec/Tat independent seems too strong. The same applies to the paragraph in the discussion section.
4. Mutant popC and popD strains are not clearly explained in the text, the reader has to go to the strain table to understand what they are. Labelling all the figures with popC, when they mean popC::aadA, is confusing. The authors should use a clearer and more intuitive nomenclature for the mutants.
5. Line 145: authors state "ectopic expression of oar at native levels". Please provide a bit more information about this ectopic expression. Is the expression driven by oar promoter (I assume that native level means this)? In a plasmid? In the genome (the wider community won't be aware of the lack of efficient propagating plasmids in *M. xanthus*)? This comment applies to all mentions of ectopic expressions in the text, they are not clearly explained.
6. Figure6: deletions of exbD1, exbD2 and exbB1 do not show a strong defect in the formation of the fruiting bodies but the effect on sporulation is very dramatic. Can the authors explain this? Can they elaborate more about the secreted popC in these mutants?
7. Fig. 8: why are OmpA and OmpX also enriched in the absence of TonB?
8. When first mentioned in the text, a UniProt accession number should be provided for PopC, since "PopC" *myxococcus* is not recognised.

Reviewer #2:

Remarks to the Author:

The secretion of PopC, a *Myxococcus xanthus* protease that plays an important role in development, has been the subject of a long-standing mystery because the protein lacks a signal peptide. In this manuscript, Gomez-Santos and co-workers present intriguing evidence that PopC is secreted by a completely novel mechanism. They show convincingly that a TonB-dependent transporter (TBDT) called Oar is required for PopC export. That finding is notable in itself because TBDTs have previously been implicated in protein import, but not protein export. On a less positive note, however, many of the authors' claims are overstated because they are only weakly supported by the data or are based entirely on speculation. I was not, for example, convinced that the secretion of PopC is driven by the proton motive force (PMF) (Fig. 2). Furthermore, the analysis of the role of the Ton(Oar) system in PopC secretion (Fig. 6) appears to be challenging to reconcile with other results and involves experiments that are tangential to the main story. Although the authors should be commended for putting a great deal of effort into trying to elucidate the mechanism of PopC secretion, the manuscript contains quite a few inconclusive or even puzzling observations. I think that the authors need to revise the manuscript considerably by including additional experiments to bolster some of the major arguments, removing tangential data, and interpreting some of their results more cautiously.

Specific comments:

1) p. 5 and Fig. 1: The authors show by Western blot that the vast majority of the PopC is located in the periplasm at steady state, but they do not provide rigorous evidence that the periplasmic form of the protein is a secretion intermediate. To do so, perhaps they could do a chloramphenicol chase, i.e., add chloramphenicol before starvation (as performed in ref. 8) and show that the level of the protein in the periplasm goes down while the level in the medium goes up over time. This sort of experiment would be useful not only because it would support the idea that PopC is secreted in a two-step process, but also because it would give a sense of the kinetics of export. Indeed information about export kinetics might ultimately provide insight into the mechanism of the export reaction. As a bonus, it would also be nice to show that the degradation of PopD precedes (or parallels) the export of PopC.

2) p. 5, bottom and Fig. 2: The evidence that the PMF drives the secretion of PopC is shaky. The data suggest that inhibitors of the PMF only partially inhibit PopC secretion (less than 50%) even after a long period of time. It is difficult to determine, however, if the PMF is required for PopC secretion but only a moderate effect was observed because (as the authors suggest) the inhibitors did not completely dissipate the PMF, or if the PMF is not required for PopC secretion but the drugs alter cell physiology in a way that indirectly impairs export. Because the results are difficult to interpret, this experiment should be removed from the manuscript, moved to the supplement, or at the very least interpreted with a great deal of caution.

3) The use of "arbitrary units" to quantitate the secretion of PopC in Figs. 2, 4, 5 and 6 seems a bit odd and should be explained. Why don't the authors quantitate secretion as "% secreted"? Is secretion inefficient? Does 1.0 "arbitrary unit" correspond to a small or large fraction of the protein secreted?

3) Several observations in Figs. 4A and 4B do not seem to make sense and should be clarified. First, is the apparent reduction in the amount of PopC in the wild-type cell extract over time due to its release into the medium, or was there simply a technical problem with the running of the gel/Western blot? If the first explanation is correct, then why is there no change in the level of PopC in the extract of any of the other strains (e.g., the popD- mutant strain). Second, if PopD inhibits PopC secretion in non-starving cells, then why isn't most of the PopC in the supernatant of the popD- strain at time t=0? Likewise, why isn't there an accumulation of p17 in the non-starving popD- cells?

4) Line 187: The data that have been presented to this point in the manuscript do not test a model "in which Oar is acting directly at the level of PopC secretion". This statement should be toned

down.

5) Lines 201-203: The authors' claim that "the observation that PopC is only detected in the membrane fraction after cross-linking suggests that [PopC and Oar] only interact transiently" is inaccurate and should be modified or removed. Exactly the opposite is true: the ability to observe a crosslink suggests that the two proteins form a relatively stable interaction. Transient interactions are more difficult to capture by crosslinking.

6) In Fig. 5 the authors show that the plug domain of Oar interacts with the CTD of TonB1, but strong point mutations in the TonB box of Oar only moderately inhibit PopC secretion (perhaps 1.5-2.5-fold). Based on this observation, it is difficult to determine if the TonB box is important for Oar function or if the point mutations reduce secretion efficiency simply because they perturb the overall structure of the plug domain. Because the results are ambiguous, the statement "We conclude that the Oar plug domain with the TonB box is important for Oar function" is an overinterpretation that should be removed or modified.

7) In Fig. 5c, the Oar(delta plug) mutant appears to be unstable. Might the instability of the mutant TBDT indicate an overall folding defect that explains the strong reduction in PopC secretion? Can the authors rule out this possibility?

8) As stated above, the analysis of the Ton(Oar) locus is rather confusing. I'm not sure how to interpret the observation that ectopic expression of tonB1 does not complement a tonB1 deletion or how that observation is relevant to the mechanism of PopC secretion. Furthermore, the finding that the deltaTon(Oar) mutation affects the localization of Oar is tangential to the main point of the study and can be omitted. A much bigger concern, however, is that the deltaTon(Oar) mutation does not affect PopC secretion (Fig. 6b). This finding seems to contradict the argument that an interaction between Oar and TonB1 drives PopC secretion. In light of the data the authors are forced to speculate that "in the absence of the Ton(Oar) system, Oar is energized by one of the remaining Ton systems" (lines 273-274), but they do not present any evidence to support their hypothesis. In light of the data, the authors need to tone down the argument that Ton systems drive PopC secretion.

9) Lines 284-286: This statement seriously overinterprets the results and should be modified. The BACTH experiment does provide evidence that the N-terminal region of Oar interacts with E. coli TonB, but that observation alone does not show that TonB energizes Oar.

10) Lines 288-291 and Fig. 7A: The solubility experiment provides very vague, ambiguous results that should probably be removed from the manuscript (or perhaps placed in the supplement). At best, the increased solubility of PopC when it is co-expressed with Oar provides weak (and indirect) evidence that the two proteins interact. Furthermore, the finding that more than half of both proteins ends up in the cell debris/inclusion body fraction confounds the interpretation of the results.

11) The cell fractionation data presented in the top two parts of Fig. 7C provide good evidence that Oar and PopC interact in E. coli, but the data presented in the bottom part really confuse the whole picture. It is not clear to me why PopC appears in the inner membrane fraction in the absence of TonB. Does the TonB deletion make the inner membrane become sticky? Or was there a technical problem with the fractionation? Unless the authors can clarify the situation, they should remove the bottom panel.

12) In Fig. 8, the authors use a highly unconventional and ambiguous method to show that PopC interacts specifically with Oar in E. coli. The quantitative proteomics methods show that they can co-immunoprecipitate Oar with PopC, but the significance of the quantitative enrichments (e.g., 4.9-fold in wild-type cells) is impossible to interpret. The authors really need to provide evidence that a significant fraction of the Oar can be immunoprecipitated with an anti-PopC antibody. If

they can't see a signal on a gel and can only obtain results using super-sensitive technology, then in my mind their conclusions are highly suspect. To complicate matters, the authors conclude that because they cannot consistently detect any native *E. coli* outer membrane proteins in their co-immunoprecipitation experiments, "no other OM protein is involved in PopC secretion and that Oar directly secretes PopC" (lines 322-323). That is a dangerous overinterpretation of negative results that needs to be modified. It is indeed possible that a key component of the export apparatus interacts transiently with PopC and cannot be detected in the co-immunoprecipitation assay.

13) Given that the authors do not clearly establish a role for the TonB system in the secretion of PopC, I wonder if PopC might bind to Oar and then be released from cells in outer membrane vesicles instead of being secreted through the middle of Oar. I raise this possibility because the authors show in Fig. 3g that they find at least some Oar protein in outer membrane vesicles. The authors should address this possibility in the manuscript.

Minor comments:

1) Line 121: To the best of my knowledge T9SSs have not been identified in proteobacteria, so this reference can be removed.

2) The Figure legends are incredibly long and filled with a lot of excess information. I suggest that the authors transfer some of the methodological details presented in the Figure legends to the Methods section so that the Figure legends are more concise and easier to read.

Reviewer #3:

Remarks to the Author:

The manuscript by Gomez-Santos et al describes a new pathway for how proteins are transported across the outer membrane (OM) of gram-negative bacteria. The cargo protein used here (PopC) is a protease involved in *M. xanthus* development. During development PopC is secreted extracellularly but how this occurs has been a puzzle because it lacks recognizable signal peptide (SP) sequences. By use of inhibitors PopC secretion was found to depend on the PMF which led these investigators to test TonB systems. The authors provide a strong experimental case that PopC is transported by a two-step mechanism and that transport across the OM is mediated by Oar, an OM TBDT protein. This finding fundamentally provides a new pathway for protein secretion because to date TBDT systems were only implicated in cargo uptake, particularly nutrients, siderophores and bacteriocins that hijacked these systems to gain cell entry. Their findings are therefore of broad interest to the microbiology field. How PopC crosses the inner membrane was not elucidated in this work and remains a mystery. Overall the text is clear and polished. However, there are a number of concerns that need to be addressed.

As presented, heterologous PopC secretion by an Oar-dependent mechanism in *E. coli* is not convincing. The key experiment is Fig. 7d (particularly top panel) where it appears (i.e. not explicitly stated) protein detection was done by western analysis. If so, it appears a very small amount (e.g. less than 1 percent) of PopC is actually secreted, because overall PopC is massively overexpressed in the cytoplasm of *E. coli* from a strong pET vector (i.e. major band in Coomassie Blue stained gel, Fig. 7a). Given this, the experimental controls (native proteins) for cell lysis are orders of magnitude less sensitive than PopC. This issue needs to be addressed by similarly overexpressing a soluble protein control and testing for its presence in the extracellular milieu by western analysis. In addition the figure legend needs to clearly describe the detection method(s) and how the supernatants were prepared/loaded onto the gel. The authors should then estimate what fraction of PopC is actually secreted plus and minus Oar expression.

It seems that the results from Fig. 8 (lines 312-326) are over interpreted, e.g. "PopC ... does not interact with other OM proteins involved in protein secretion." The authors make the point that Oar

was enriched 4.9 fold in the presence of PopC (IP expts), but from Fig. 8b it is also shown, for example, that OmpX is enriched 8.6-fold in the presence of PopC. How can the former result be significant and the latter not be significant?

Result section: A prior study by Bhat, Zhu, Patel, Orlando and Shimkets (2011) already showed that Oar is required for C-signaling and therefore this work should be properly cited.

Experiments were done in an atypical WT background of *M. xanthus* (DK101). Because of this the authors should describe/justify why this background was used. Second, for clarity, in Table S2, DK101 should be designated as "WT."

Minor points:

Fig. 4c. In the western blot (top), please explain why is the Oar band is more intense when the WT sample is treated with DSP (minus DTT) compared to the other WT/popC samples?

Fig. 4a, c and legend. The agIB1 mutation in DK1217 is allelic with agIQ (Dey et al., 2016 J. Bacteriol. 198(6):994, Supplemental text); however this point is not described and creates confusion.

Lines 64-5. To aid the reader describe what is known about the sub-cellular localization of PopD.

Lines 102-4. Explain why chloramphenicol treatment before starvation demonstrates that PopC is secreted by a two-step mechanism (not obvious).

Line 114. Insert "and" after "reversibly"

Line 165-6: For clarity a sentence should be added to explain the epistasis results; i.e. why are these genes in the same pathway.

Line 168-9: Although the oar mutation is complemented for PopC secretion there is nevertheless an accumulation of PopC in the cell extract which is not seen in WT (Fig. 4a). This point should be mentioned in the text.

Line 172: Section heading is a bit awkward, change to, for example, "Oar interacts with PopC and has a direct role in its secretion"

Line 179: insert "the" after "in"

Lines 216-7: To reflect western findings shown in Fig. 5c, after "OM" insert ", although to a lower degree than WT." In addition, reflecting this accumulation caveat, in line 219 change "essential" to "important"

Line 252. Given that the tonB1 gene lies upstream of oar, in an opposite orientation, and that the deletion of tonB1 results in no oar expression, the authors should state the bp distance from the predicted oar start codon to the boundary of the tonB1 deletion (i.e. was the oar promoter deleted?).

Line 258. For clarity insert "heterologous" or "fortuitous" before "promoter" if this is what the authors think occurred.

Line 279. Change "required" to "necessary"

Line 283-6. Run on sentence, add commas and/or break into two sentences.

Line 993. Capitalize "table S3."

Reviewers' comments:

Reviewer #1 (Remarks to the Author):

The manuscript by Nuria-Gomez et al describes the discovery of a TonB-dependent transporter (TBDT Oar) that functions as a protein secretion channel for the protease PopC. This is surprising and quite exciting because so far, TBDTs are known to be exclusively involved in OM uptake processes. The evidence presented for this is quite (almost too) elaborate and overall very convincing. If anything, the relevance of this important finding could be presented a bit more strongly, for example in the abstract, where the discovery of a novel secretion system is mentioned almost as an afterthought. As is, the study will form the basis for more detailed investigations into the mechanism of secretion by Oar. Like I said, this is a very nice paper and I only have some minor suggestions for clarification and improvement.

Response: Thank very much! We have slightly rewritten the Abstract to emphasize that we present a novel secretion system and now write “identify a novel mechanism for protein secretion to the extracellular milieu that depends on a TonB-dependent transporter (TBDT) and the ExbB/ExbD/TonB system”.

1. Is it possible to say how widespread the Oar TBDT secretion system is based on sequence conservation and perhaps phylogenetic analysis? Is it unique to *Myxococcus* spp? Does the Oar sequence have features that set it apart from regular "import" TBDTs?

Response: To address this question we included in the revised manuscript a novel bioinformatics analysis of TBDTs. Briefly, we find that TBDTs are ubiquitous and can be divided into five distinct groups based on the N-terminal domain architecture. Moreover, we find that the taxonomic distribution of proteins with different architectures is highly biased (line 383-395; Suppl. Fig. 8; Suppl. Tables 2 and 4).

We also discuss that the two variants of Oar that secrete PopC, i.e. the WT protein with the N-terminal extension as well as the variant without this extension, represent the two architectures that account for 96% of all TBDTs. Interestingly, proteins with this domain architecture have also been shown to support import processes. Therefore, we write that “Thus, the sequence features that may distinguish a TBDT involved in secretion from one involved in import are not currently known and their identification awaits the characterization of additional TBDTs involved in protein secretion. However, based on the data presented here, we speculate that some of the many uncharacterized TBDTs may not only be involved in import but additionally, or exclusively, in protein secretion across the OM” (line 465-478).

2. The authors should clarify with wt strain of *M. xanthus* they are using. Although it is stated in Material and method, adding this information to the main text would help the reader.

Response: Done (line 99) and also included in Methods and in Suppl. Table 3.

3. Line 93: the authors state “because PopC is a soluble protein, lacks a signal peptide and is secreted as a full-length”. They cite their own work (reference number 7). In that study, SignalP and TatP programs are used for an in silico search for a potential signal sequence. Authors claim that because no signal sequence is found by this software there is no signal sequence. This statement is also based in SDS and Western blots of PopC. However, signal sequences can be short, and a ~2 kDa difference might be difficult to see on a SDS gel. If the first ~20 amino acids are removed, is PopC still moved to the periplasm in *E. coli*? This

would be simple to test. As it is, the statement that PopC secretion is Sec/Tat independent seems too strong. The same applies to the paragraph in the discussion section.

Response: Thanks for pointing this out to us. To determine precisely whether PopC is secreted as a full-length protein, we performed an additional experiment. In this experiment, we determined the N-terminus of PopC from the cell-free supernatant as well as in total cell extract from 6 h starving cells using mass spectrometry. As described in line 103-107 and Supp. Fig. 1b, the native PopC N-terminus was detected in both fractions. Therefore, we conclude that PopC is secreted to the extracellular milieu as a full-length protein.

4. Mutant popC and popD strains are not clearly explained in the text, the reader has to go to the strain table to understand what they are. Labelling all the figures with popC, when they mean popC::aadA, is confusing. The authors should use a clearer and more intuitive nomenclature for the mutants.

Response: We apologize for the confusion. We have now included in all figures the correct nomenclature for the alleles. Moreover, we write in the legend to Fig. 1 that “Note that throughout the text, the *popC::aadA* allele is referred to as *popC*”. Also, in the legend to Fig. 4, we write that “Note that throughout the text, the *popD::aadA* allele is referred to as *popD*”.

5. Line 145: authors state “ectopic expression of oar at native levels”. Please provide a bit more information about this ectopic expression. Is the expression driven by oar promoter (I assume that native level means this)? In a plasmid? In the genome (the wider community won't be aware of the lack of efficient propagating plasmids in *M. xanthus*)? This comment applies to all mentions of ectopic expressions in the text, they are not clearly explained.

Response: We apologize for the lack of clarity. We have included throughout the text that genes for complementation are expressed ectopically from the *pilA* promoter. Moreover, we included in Methods (line 491-492) that “Plasmids for ectopic expression of genes in *M. xanthus* were integrated by site specific recombination at the *attB* site and the relevant genes expressed from the *pilA* promoter”. Referring to the ectopic expression of *oar*, we now write “in a Δoar strain in which *oar* was ectopically expressed from the *pilA* promoter on a plasmid integrated in the genome at the *attB* site, Oar accumulated at the same level as in WT (Fig. 3f, lower)” (line 174-175).

6. Figure 6: deletions of *exbD1*, *exbD2* and *exbB1* do not show a strong defect in the formation of the fruiting bodies but the effect on sporulation is very dramatic. Can the authors explain this? Can they elaborate more about the secreted popC in these mutants?

Response: To clarify the phenotype of the $\Delta exbD2$, $\Delta exbD1$ and $\Delta exbB1$ mutants, we have rewritten the text as follows “Analyses of these mutants demonstrated that after 120 h of starvation the $\Delta exbD2$, $\Delta exbD1$ and $\Delta exbB1$ mutants had a developmental phenotype similar to that of the Δoar mutant whereas the $\Delta tonB1$ mutant had a more severe defect in fruiting body formation” (line 278-280).

Concerning PopC secretion in these mutants we have rewritten to “Surprisingly, the $\Delta exbD2$, $\Delta exbD1$ and $\Delta exbB1$ mutants still secreted PopC albeit at a significantly lower level than in WT (Fig. 6b)” (line 281-282) and “The Δton^{oar} mutant had developmental defects (Fig. 6a), accumulated PopC in total cell extracts and secreted PopC in an Oar-dependent manner at WT levels (Fig. 6b). Surprisingly, however, the Δton^{oar} mutant did not accumulate p17 while p25 accumulated at WT levels (Fig. 6f)” (line 306-309). These paradoxical observations, i.e.

(i) PopC is secreted in Ton system mutants despite the fact that the plug domain in Oar and the TonB box are important for PopC secretion, and (ii) PopC is secreted by the Δton^{oar} mutant but p25 is not processed to p17, led us to do follow up experiments as follows: (a) We speculated that Oar could be energized by one of the remaining Ton systems in *M. xanthus* (Fig. 3a). To test this idea we performed additional experiments in which we now demonstrate that the N-terminal domains of Oar including the plug domain with the TonB box interact with the CTD of the TonB proteins MXAN_0276 and MXAN_0820 in BACTH analysis (Fig. 3a; Supplementary Fig. 6) (line 311-317). (b) We demonstrate that the Ton^{Oar} system is important for polar localization of Oar. Therefore, “We speculate that in the absence of the Ton^{Oar} system, Oar is energized by one of the remaining Ton systems; however, in this situation, PopC would be secreted away from its substrate (Fig. 6h). Because PopC has a short half-life after secretion to the extracellular milieu⁷, PopC secreted away from its substrate may not efficiently cleave p25 and, thus, p17 would not accumulate” (line 328-332).

7. Fig. 8: why are OmpA and OmpX also enriched in the absence of TonB?

Response: To clarify the co-IP experiments with α -PopC antibodies on *E. coli* membranes isolated from WT and the $\Delta tonB$ mutant expressing PopC and/or Oar, we have now included new experiments in which we use immunoblotting to show that Oar is immunoprecipitated in a PopC-dependent manner. Moreover, we quantified the co-IP experiments and demonstrate that Oar was enriched 4.9-fold in WT and 15.1-fold in the $\Delta tonB$ mutant in the presence of PopC (line 372-381; Fig. 8a, b). We decided not to include the data on the other much less enriched outer membrane proteins identified in the co-IP experiments using label-free quantitative mass spectrometry in order to keep focus on the important observation that Oar is enriched in the presence of PopC in these experiments.

8. When first mentioned in the text, a UniProt accession number should be provided for PopC, since "PopC" myxococcus is not recognised.

Response: Done (line 61).

Reviewer #2 (Remarks to the Author):

The secretion of PopC, a *Myxococcus xanthus* protease that plays an important role in development, has been the subject of a long-standing mystery because the protein lacks a signal peptide. In this manuscript, Gomez-Santos and co-workers present intriguing evidence that PopC is secreted by a completely novel mechanism. They show convincingly that a TonB-dependent transporter (TBDT) called Oar is required for PopC export. That finding is notable in itself because TBDTs have previously been implicated in protein import, but not protein export. On a less positive note, however, many of the authors' claims are overstated because they are only weakly supported by the data or are based entirely on speculation. I was not, for example, convinced that the secretion of PopC is driven by the proton motive force (PMF) (Fig. 2). Furthermore, the analysis of the role of the Ton(Oar) system in PopC secretion (Fig. 6) appears to be challenging to reconcile with other results and involves experiments that are tangential to the main story. Although the authors should be commended for putting a great deal of effort into trying to elucidate the mechanism of PopC secretion, the manuscript contains quite a few inconclusive or even puzzling observations. I think that the authors need to revise the manuscript considerably by including additional experiments to bolster some of the major arguments, removing tangential data, and

interpreting some of their results more cautiously.

Response: Thanks for the positive feedback. Regarding the more critical comments, please refer to our answers below.

Specific comments:

1) p. 5 and Fig. 1: The authors show by Western blot that the vast majority of the PopC is located in the periplasm at steady state, but they do not provide rigorous evidence that the periplasmic form of the protein is a secretion intermediate. To do so, perhaps they could do a chloramphenicol chase, i.e., add chloramphenicol before starvation (as performed in ref. 8) and show that the level of the protein in the periplasm goes down while the level in the medium goes up over time. This sort of experiment would be useful not only because it would support the idea that PopC is secreted in a two-step process, but also because it would give a sense of the kinetics of export. Indeed information about export kinetics might ultimately provide insight into the mechanism of the export reaction. As a bonus, it would also be nice to show that the degradation of PopD precedes (or parallels) the export of PopC.

Response: Thanks for suggesting the chloramphenicol chase experiment! We have now included this experiment in Supp. Fig. 1c and it is described in line 120-129. This new experiment clearly shows that over the course of the chloramphenicol treatment of starving cells, the PopC level in the periplasmic fractions decreased; by contrast, the PopC level in the cytoplasmic fractions did not decrease. This observation (together with the finding that PopC is strongly enriched in the periplasm) support that PopC is secreted to the extracellular milieu from the periplasm suggesting that PopC secretion to the extracellular milieu occurs in a two-step mechanism.

Concerning PopD degradation: We have previously shown that PopC and PopD interact when co-expressed in *E. coli* (Konovalova *et al.* 2012) and that PopD is an inhibitor of PopC secretion in *M. xanthus* (Konovalova *et al.* 2012). Unfortunately, we are unable to detect PopD by immunoblotting in *M. xanthus*. We have used several different strategies to obtain antibodies against PopD but without success (antibodies against full-length PopD as well as peptide antibodies). Moreover, we have epitope-tagged PopD with a His6-tag and a Strep-tag in order to use commercial antibodies. Again, we cannot detect PopD in immunoblots. Therefore, we do not know to which subcellular compartment PopD localizes in *M. xanthus*. However, in co-immunoprecipitation experiments in *M. xanthus* with α -PopC antibodies, PopD is enriched. So, we are confident that PopC and PopD also interact in *M. xanthus*. Preliminary data from experiments in *E. coli* in which PopC and PopD are co-expressed suggest that there are two pools of PopC: a less abundant cytoplasmic pool in which PopC is in a complex with PopD and a more abundant periplasmic pool in which PopC is not in a complex with PopD. So, if we extrapolate these findings to *M. xanthus*, they suggest that PopD is in the cytoplasm in a complex with PopC. We would like to add that PopC is also detected in the cytoplasmic fraction of non-starving and starving *M. xanthus* cells (Fig. 1a, b) as well as in *E. coli* cells (Fig. 7b). We are currently in the process of untangling the PopC/PopD interactions and how these interactions allow PopD to function as an inhibitor of PopC secretion. We would like to add that the open questions regarding PopD do not affect the conclusion that PopC is secreted by Oar in a manner that depends on the Ton system in *M. xanthus* as well as in *E. coli*.

2) p. 5, bottom and Fig. 2: The evidence that the PMF drives the secretion of PopC is shaky. The data suggest that inhibitors of the PMF only partially inhibit PopC secretion (less than

50%) even after a long period of time. It is difficult to determine, however, if the PMF is required for PopC secretion but only a moderate effect was observed because (as the authors suggest) the inhibitors did not completely dissipate the PMF, or if the PMF is not required for PopC secretion but the drugs alter cell physiology in a way that indirectly impairs export. Because the results are difficult to interpret, this experiment should be removed from the manuscript, moved to the supplement, or at the very least interpreted with a great deal of caution.

Response: We apologize for the confusion. In the case of *M. xanthus*, 10 μ M and 100 μ M CCCP and nigericin, respectively have been reported to dissipate the PMF and the proton gradient, respectively (Sun *et al.* 2011). However, in our hands concentrations higher than 5 μ M CCCP and 50 μ M nigericin cause cell lysis. Therefore, we originally used 1 or 5 μ M CCCP and 10 or 50 μ M nigericin in our experiments and, consequently, the PMF and proton gradient may not be completely dissipated. To present our data more clearly, we modified Fig. 2 and now (i) only include the data with 5 μ M CCCP and 50 μ M nigericin. Moreover, we have changed the way in which we present the data to emphasize that (a) PopC secretion is significantly reduced in cells treated with CCCP and nigericin compared to untreated cells within 30 min, and (ii) PopC levels in the cell-free supernatant do not change significantly after 10 min of exposure to CCCP and nigericin. In total, these observations suggest that the PMF and the pH gradient are important for PopC secretion across the OM (line 132-146). As part of these modifications, we also moved all measurements of ATP levels to Suppl. Fig. 2. Therefore, we find that the statement “We conclude that the PMF and the pH gradient are important for PopC secretion across the OM” (line 145-146) is valid.

3) The use of “arbitrary units” to quantitate the secretion of PopC in Figs. 2, 4, 5 and 6 seems a bit odd and should be explained. Why don't the authors quantitate secretion as “% secreted”? Is secretion inefficient? Does 1.0 “arbitrary unit” correspond to a small or large fraction of the protein secreted?

Response: We have now included this information in the legend to Fig. 4 and state that “1 a.u. corresponds to $3 \pm 1\%$ of PopC in total cell extracts”.

3) Several observations in Figs. 4A and 4B do not seem to make sense and should be clarified. First, is the apparent reduction in the amount of PopC in the wild-type cell extract over time due to its release into the medium, or was there simply a technical problem with the running of the gel/Western blot? If the first explanation is correct, then why is there no change in the level of PopC in the extract of any of the other strains (e.g., the *popD*- mutant strain). Second, if PopD inhibits PopC secretion in non-starving cells, then why isn't most of the PopC in the supernatant of the *popD*- strain at time $t=0$? Likewise, why isn't there an accumulation of p17 in the non-starving *popD*- cells?

Response: We apologize for this confusion! There was indeed a technical problem with the blot shown for the WT strain in Fig. 4a. Therefore, we exchanged this blot with a technical replicate, which clearly shows that the amount of PopC in total cell extracts only decreases slightly during starvation. Concerning the *popD* mutant: There is statistically, significantly more PopC in the cell-free supernatant of the *popD* mutant than in the WT at $t=0$. This difference is not very large; but please note that the difference in PopC secretion during starvation in the presence and absence of PopD is also not very large. Concerning the formation of p17 in non-starving *popD* cells: The immunoblot does show a p17 band. This band is relatively weak consistent with the low level of PopC secretion under these

conditions.

4) Line 187: The data that have been presented to this point in the manuscript do not test a model “in which Oar is acting directly at the level of PopC secretion”. This statement should be toned down.

Response: We have rephrased this sentence to “...and support a model in which Oar may be acting at the level of PopC secretion” (line 227-228).

5) Lines 201-203: The authors’ claim that “the observation that PopC is only detected in the membrane fraction after cross-linking suggests that [PopC and Oar] only interact transiently” is inaccurate and should be modified or removed. Exactly the opposite is true: the ability to observe a crosslink suggests that the two proteins form a relatively stable interaction. Transient interactions are more difficult to capture by crosslinking.

Response: We have rephrased this statement to “...also suggests that the two proteins may only interact transiently” (line 244).

6) In Fig. 5 the authors show that the plug domain of Oar interacts with the CTD of TonB1, but strong point mutations in the TonB box of Oar only moderately inhibit PopC secretion (perhaps 1.5-2.5-fold). Based on this observation, it is difficult to determine if the TonB box is important for Oar function or if the point mutations reduce secretion efficiency simply because they perturb the overall structure of the plug domain. Because the results are ambiguous, the statement “We conclude that the Oar plug domain with the TonB box is important for Oar function” is an overinterpretation that should be removed or modified.

Response: In the experiments in Fig. 5c and 5d, we show that an Oar variant lacking the plug domain accumulated at a lower level than the WT protein in total cell extracts but still accumulated in the outer membrane at WT level. This variant is significantly reduced in supporting PopC secretion. Moreover, the two Oar variants with single amino acid substitutions of conserved residues in the TonB box accumulate at WT levels and still accumulate in the outer membrane. These two Oar variants are also significantly reduced in supporting PopC secretion. Importantly, substitutions of the corresponding conserved residues in other TBDTs have different effects in different proteins varying from completely blocking activity to a slight reduction of activity (Braun & Endriss, 2007). Based on these considerations, we believe that the conclusion “the Oar plug domain with the TonB box is important for Oar function and PopC secretion, most likely by directly interacting with the CTD of TonB1” (line 273-274) is justified.

7) In Fig. 5c, the Oar(delta plug) mutant appears to be unstable. Might the instability of the mutant TBDT indicate an overall folding defect that explains the strong reduction in PopC secretion? Can the authors rule out this possibility?

Response: The Oar variant that lack the plug domain accumulated at a lower level than the WT protein in total cell extracts. However, this variant still accumulated in the outer membrane. The observation that this protein is integrated in the outer membrane argues that it is correctly folded. We have rephrased the description of this Oar variant to “...accumulated at a lower level than the WT protein in total cell extracts but still accumulated in the OM at WT levels” (line 258-259).

8) As stated above, the analysis of the Ton(Oar) locus is rather confusing. I'm not sure how to interpret the observation that ectopic expression of tonB1 does not complement a tonB1 deletion or how that observation is relevant to the mechanism of PopC secretion.

Furthermore, the finding that the deltaTon(Oar) mutation affects the localization of Oar is tangential to the main point of the study and can be omitted. A much bigger concern, however, is that the deltaTon(Oar) mutation does not affect PopC secretion (Fig. 6b). This finding seems to contradict the argument that an interaction between Oar and TonB1 drives PopC secretion. In light of the data the authors are forced to speculate that "in the absence of the Ton(Oar) system, Oar is energized by one of the remaining Ton systems" (lines 273-274), but they do not present any evidence to support their hypothesis. In light of the data, the authors need to tone down the argument that Ton systems drive PopC secretion.

Response: To understand whether Oar function depends on the Ton^{Oar} system we consider that it is essential to analyse the effect of in-frame deletions in each of the four individual genes in this gene cluster. These analyses demonstrate that all four proteins are important for development and to a lesser degree for PopC secretion. In the light of these results, we went ahead and generated the $\Delta tonB1-exbD2$ quadruple mutant (Δton^{oar} mutant). This mutant also had developmental defects and still secreted PopC but did not process p25 to generate p17 (we did not do the latter analysis on the four single gene deletions).

These paradoxical observations, i.e. (i) PopC is secreted in Ton system mutants despite the fact that the plug domain in Oar and the TonB box are important for PopC secretion, and (ii) PopC is secreted by the Δton^{oar} mutant but p25 is not processed to p17, led us to do additional follow up experiments as follows: (a) We speculated that Oar could be energized by one of the remaining Ton systems in *M. xanthus*. To test this idea we have now performed additional experiments in which we demonstrate that the N-terminal domains of Oar including the plug domain with the TonB box interact with the CTD of the TonB proteins MXAN_0276 and MXAN_0820 in BACTH analysis (Fig. 3a; Supplementary Fig. 6) (line 312-318). (b) We demonstrate that the Ton^{Oar} system is important for polar localization of Oar. Therefore, "We speculate that in the absence of the Ton^{Oar} system, Oar is energized by one of the remaining Ton systems; however, in this situation, PopC would be secreted away from its substrate (Fig. 6h). Because PopC has a short half-life after secretion to the extracellular milieu, PopC secreted away from its substrate may not efficiently cleave p25 and, thus, p17 would not accumulate" (line 329-333). We find that these conclusions are justified by the data provided. In this context, we would also like to add that PopC secretion by *E. coli* is not only Oar-dependent but also depends on an intact Ton system (Fig. 7 and Suppl. Fig 7a, b). We do not agree that the effect of the Ton^{Oar} system on Oar localization is tangential. These data establish a clear connection between the Ton^{Oar} system and Oar in PopC secretion.

9) Lines 284-286: This statement seriously overinterprets the results and should be modified. The BACTH experiment does provide evidence that the N-terminal region of Oar interacts with *E. coli* TonB, but that observation alone does not show that TonB energizes Oar.

Response: We have deleted this sentence (line 341).

10) Lines 288-291 and Fig. 7A: The solubility experiment provides very vague, ambiguous results that should probably be removed from the manuscript (or perhaps placed in the supplement). At best, the increased solubility of PopC when it is co-expressed with Oar provides weak (and indirect) evidence that the two proteins interact. Furthermore, the finding

that more than half of both proteins ends up in the cell debris/inclusion body fraction confounds the interpretation of the results.

Response: We find that it is important to have the three cell fractions of *E. coli* cells expressing PopC, Oar and/or MXAN_0272 presented next to each other for ease of comparison. We agree that the finding that Oar helps to increase the solubility of PopC is indirect evidence that the two proteins interact. Importantly, several other lines of evidence support that the two proteins interact directly: (i) Oar and PopC can be cross-linked resulting in the association of PopC with the membrane fraction in *M. xanthus*. (ii) PopC associated with the OM in *E. coli* when co-expressed with Oar. (iii) Oar was enriched in co-IP experiments using α -PopC antibodies in *E. coli* cells expressing Oar together with PopC compared to *E. coli* cells only expressing Oar. In total, we would like to keep the three cell fractions of *E. coli* cells expressing PopC, Oar and/or MXAN_0272 presented next to each other for ease of comparison and we do not conclude that PopC and Oar interact solely based on the solubility effect of Oar on PopC.

11) The cell fractionation data presented in the top two parts of Fig. 7C provide good evidence that Oar and PopC interact in *E. coli*, but the data presented in the bottom part really confuse the whole picture. It is not clear to me why PopC appears in the inner membrane fraction in the absence of TonB. Does the TonB deletion make the inner membrane become sticky? Or was there a technical problem with the fractionation? Unless the authors can clarify the situation, they should remove the bottom panel.

Response: The important conclusion from Fig. 7c is that Oar is in the outer membrane and that PopC associates with the outer membrane in an Oar-dependent manner in *E. coli* WT and in the *E. coli tonB* mutant. Why more of PopC associates with the IM in the absence of TonB is not clear. However, “we speculate that the differences observed in PopC association with the OM in *E. coli* compared to *M. xanthus* are caused by higher levels of PopC and Oar accumulation in *E. coli* leading to saturation of the Ton system in *E. coli* and detection of translocation intermediates” (line 355-358). We consider that it is important to show these data because the *tonB* mutant does not secrete PopC to the cell-free supernatant albeit PopC reaches the OM in this mutant.

12) In Fig. 8, the authors use a highly unconventional and ambiguous method to show that PopC interacts specifically with Oar in *E. coli*. The quantitative proteomics methods show that they can co-immunoprecipitate Oar with PopC, but the significance of the quantitative enrichments (e.g., 4.9-fold in wild-type cells) is impossible to interpret. The authors really need to provide evidence that a significant fraction of the Oar can be immunoprecipitated with an anti-PopC antibody. If they can't see a signal on a gel and can only obtain results using super-sensitive technology, then in my mind their conclusions are highly suspect. To complicate matters, the authors conclude that because they cannot consistently detect any native *E. coli* outer membrane proteins in their co-immunoprecipitation experiments, “no other OM protein is involved in PopC secretion and that Oar directly secretes PopC” (lines 322-323). That is a dangerous overinterpretation of negative results that needs to be modified. It is indeed possible that a key component of the export apparatus interacts transiently with PopC and cannot be detected in the co-immunoprecipitation assay.

Response: To clarify the co-IP experiments with α -PopC antibodies on *E. coli* membranes isolated from WT and the $\Delta tonB$ mutant expressing PopC and/or Oar, we have now included new experiments in which we use immunoblotting to show that Oar is immunoprecipitated in

a PopC-dependent manner. Moreover, we quantified the co-IP experiments using label-free quantitative mass spectrometry and demonstrate that Oar was enriched 4.9-fold in WT and 15.1-fold in the $\Delta tonB$ mutant in the presence of PopC (line 373-382; Fig. 8a, b). We decided not to include data on the other much less enriched outer membrane proteins identified in the co-IP experiments using label-free quantitative mass spectrometry in order to keep focus on the important observation that Oar is enriched in the presence of PopC in these experiments.

13) Given that the authors do not clearly establish a role for the TonB system in the secretion of PopC, I wonder if PopC might bind to Oar and then be released from cells in outer membrane vesicles instead of being secreted through the middle of Oar. I raise this possibility because the authors show in Fig. 3g that they find at least some Oar protein in outer membrane vesicles. The authors should address this possibility in the manuscript.

Response: Thank you for suggesting this experiment! We have now included a new experiment in Suppl. Fig 1a in which we specifically address whether PopC is present in outer membrane vesicles. As shown in Suppl. Fig. 1a and described in line 97-101, Oar is detected in outer membrane vesicles but PopC is not; conversely, Oar is not detected in the cell-free supernatant but PopC is.

Minor comments:

1) Line 121: To the best of my knowledge T9SSs have not been identified in proteobacteria, so this reference can be removed.

Response: We agree that T9SSs have not been identified in proteobacteria. However, for the sake of completion and to aid the reader, we would like to keep the statement that *M. xanthus* does not encode a T9SS.

2) The Figure legends are incredibly long and filled with a lot of excess information. I suggest that the authors transfer some of the methodological details presented in the Figure legends to the Methods section so that the Figure legends are more concise and easier to read.

Response: We apologize for this. We have shortened and streamlined the legends throughout and moved experimental details to the Methods section.

Reviewer #3 (Remarks to the Author):

The manuscript by Gomez-Santos et al describes a new pathway for how proteins are transported across the outer membrane (OM) of gram-negative bacteria. The cargo protein used here (PopC) is a protease involved in *M. xanthus* development. During development PopC is secreted extracellularly but how this occurs has been a puzzle because it lacks recognizable signal peptide (SP) sequences. By use of inhibitors PopC secretion was found to depend on the PMF which led these investigators to test TonB systems. The authors provide a strong experimental case that PopC is transported by a two-step mechanism and that transport across the OM is mediated by Oar, an OM TBDT protein. This finding fundamentally provides a new pathway for protein secretion because to date TBDT systems were only implicated in cargo uptake, particularly nutrients, siderophores and bacteriocins that hijacked these systems to gain cell entry. Their findings are therefore of broad interest to the microbiology field. How PopC crosses the inner membrane was not elucidated in this work and remains a mystery. Overall the text is clear and polished. However, there are a

number of concerns that need to be addressed.

Response: Thank you!

As presented, heterologous PopC secretion by an Oar-dependent mechanism in *E. coli* is not convincing. The key experiment is Fig. 7d (particularly top panel) where it appears (i.e. not explicitly stated) protein detection was done by western analysis. If so, it appears a very small amount (e.g. less than 1 percent) of PopC is actually secreted, because overall PopC is massively overexpressed in the cytoplasm of *E. coli* from a strong pET vector (i.e. major band in Coomassie Blue stained gel, Fig. 7a). Given this, the experimental controls (native proteins) for cell lysis are orders of magnitude less sensitive than PopC. This issue needs to be addressed by similarly overexpressing a soluble protein control and testing for its presence in the extracellular milieu by western analysis. In addition the figure legend needs to clearly describe the detection method(s) and how the supernatants were prepared/loaded onto the gel. The authors should then estimate what fraction of PopC is actually secreted plus and minus Oar expression.

Response: Thanks for suggesting overexpressing a soluble protein! We followed the advice of the reviewer and overexpressed in the WT *E. coli* strain two variants of the MalE protein, cMalE and pMalE that are targeted to the cytoplasm and periplasm, respectively. As shown in Suppl. Fig. 7c-e and described in line 363-368, none of the two proteins are detected in the cell-free supernatant. Therefore, we conclude "...that the presence of PopC in the cell-free supernatant of the WT *E. coli* strain with an intact Ton system and co-expressing Oar and PopC, is the result of *bona fide* secretion of PopC" (line 368-370).

Concerning the figure legends: We apologize for not being sufficiently clear. We have now amended all legends to make experimental details clear.

Concerning the amount of PopC that is secreted to the cell-free supernatant we have now included in the legend of Fig. 7d that "PopC detected in the cell-free supernatant corresponds to ~0.2 % of PopC in cleared cell lysates".

It seems that the results from Fig. 8 (lines 312-326) are over interpreted, e.g. "PopC ... does not interact with other OM proteins involved in protein secretion." The authors make the point that Oar was enriched 4.9 fold in the presence of PopC (IP expts), but from Fig. 8b it is also shown, for example, that OmpX is enriched 8.6-fold in the presence of PopC. How can the former result be significant and the latter not be significant?

Response: To clarify the co-IP experiments with α -PopC antibodies on *E. coli* membranes isolated from WT and the $\Delta tonB$ mutant expressing PopC and/or Oar, we have now included new experiments in which we use immunoblotting to show that Oar is immunoprecipitated in a PopC-dependent manner. Moreover, we quantified the co-IP experiments using label-free quantitative mass spectrometry and demonstrate that Oar was enriched 4.9-fold in WT and 15.1-fold in the $\Delta tonB$ mutant in the presence of PopC (line 373-382; Fig. 8a, b). We decided not to include data on the other much less enriched outer membrane proteins identified in the co-IP experiments using label-free quantitative mass spectrometry in order to keep focus on the important observation that Oar is enriched in the presence of PopC in these experiments.

Result section: A prior study by Bhat, Zhu, Patel, Orlando and Shimkets (2011) already showed that Oar is required for C-signaling and therefore this work should be properly cited.

Response: Thanks & done (line 169-170).

Experiments were done in an atypical WT background of *M. xanthus* (DK101). Because of this the authors should describe/justify why this background was used. Second, for clarity, in Table S2, DK101 should be designated as “WT.”

Response: We do the *M. xanthus* experiments in the DK101 wild-type strain because this wild-type strain is the only strain in which a deletion of the *relA* gene has been reported to be successfully obtained. Because RelA is essential for PopC secretion (Konovalova *et al.* 2012), we had to use this wild-type strain.

Minor points:

Fig. 4c. In the western blot (top), please explain why is the Oar band is more intense when the WT sample is treated with DSP (minus DTT) compared to the other WT/popC samples?

Response: This signal corresponds to the crosslinked Oar-PopC product. It is commonly observed that cross-linked proteins give an enhanced signal in western blots (e.g. Akita *et al.* J Cell Biology 136: 983-994 (1997); Kota and Ljungdahl. J Cell Biology 168: 79-88 (2005)).

Fig. 4a, c and legend. The agIB1 mutation in DK1217 is allelic with agIQ (Dey *et al.*, 2016 J. Bacteriol. 198(6):994, Supplemental text); however this point is not described and creates confusion.

Response: Thanks! This information has now been included in the strain table (=Suppl. Table 3).

Lines 64-5. To aid the reader describe what is known about the sub-cellular localization of PopD.

Response: We have previously shown that PopC and PopD interact when co-expressed in *E. coli* (Konovalova *et al.* 2012) and that PopD is an inhibitor of PopC secretion in *M. xanthus* (Konovalova *et al.* 2012). Unfortunately, we are unable to detect PopD by immunoblotting in *M. xanthus*. We have used several different strategies to obtain antibodies against PopD but without success (antibodies against full-length PopD as well as peptide antibodies). Moreover, we have epitope-tagged PopD with a His6-tag and a Strep-tag in order to use commercial antibodies. Again, we cannot detect PopD in immunoblots. Therefore, we do not know to which subcellular compartment PopD localizes in *M. xanthus*. However, in co-immunoprecipitation experiments in *M. xanthus* with α -PopC antibodies, PopD is enriched. So, we are confident that PopC and PopD also interact in *M. xanthus*. Preliminary data from experiments in *E. coli* in which PopC and PopD are co-expressed suggest that there are two pools of PopC: a less abundant cytoplasmic pool in which PopC is in a complex with PopD and a more abundant periplasmic pool in which PopC is not in a complex with PopD. So, if we extrapolate these findings to *M. xanthus*, they suggest that PopD is in the cytoplasm in a complex with PopC. We would like to add that PopC is also detected in the cytoplasmic fraction of non-starving and starving *M. xanthus* cells (Fig. 1a, b) as well as in *E. coli* cells (Fig. 7b). We are currently in the process of untangling the PopC/PopD interactions and how these interactions allow PopD to function as an inhibitor of PopC secretion. We would like to add that the open questions regarding PopD do not affect the conclusion that PopC is secreted by Oar in a manner that depends on a Ton system in *M. xanthus* as well as in *E.*

coli.

Lines 102-4. Explain why chloramphenicol treatment before starvation demonstrates that PopC is secreted by a two-step mechanism (not obvious).

Response: As suggested by reviewer 1, we did a chloramphenicol chase experiment. We have now included this experiment in Supp. Fig. 1c and it is described in line 120-129. This new experiment clearly shows that over the course of the chloramphenicol treatment of starving cells, the PopC level in the periplasmic fractions decreased; by contrast, the PopC level in the cytoplasmic fractions did not decrease. This observation (together with the finding that PopC is strongly enriched in the periplasm) support that PopC is secreted to the extracellular milieu from the periplasm suggesting that PopC secretion to the extracellular milieu occurs in a two-step mechanism.

Line 114. Insert “and” after “reversibly”

Response: Thanks & done (line 141).

Line 165-6: For clarity a sentence should be added to explain the epistasis results; i.e. why are these genes in the same pathway.

Response: We have now included this information in line 195-200.

Line 168-9: Although the *oar* mutation is complemented for PopC secretion there is nevertheless an accumulation of PopC in the cell extract which is not seen in WT (Fig. 4a). This point should be mentioned in the text.

Response: We apologize for this confusion! There was a technical problem with the blot shown for the WT strain in Fig. 4a. Therefore, we exchanged this blot with a technical replicate, which clearly show that the amount of PopC in total cell extracts only decreases slightly during starvation similarly to what is observed for the complemented *oar* mutant.

Line 172: Section heading is a bit awkward, change to, for example, “Oar interacts with PopC and has a direct role in its secretion”

Response: Thanks & done (line 213).

Line 179: insert “the” after “in”

Response: Thanks & done (line 219).

Lines 216-7: To reflect western findings shown in Fig. 5c, after “OM” insert “, although to a lower degree than WT.” In addition, reflecting this accumulation caveat, in line 219 change “essential” to “important”

Response: Thanks & rewritten (line 258 and 262).

Line 252. Given that the *tonB1* gene lies upstream of *oar*, in an opposite orientation, and that the deletion of *tonB1* results in no *oar* expression, the authors should state the bp distance from the predicted *oar* start codon to the boundary of the *tonB1* deletion (i.e. was the *oar* promoter deleted?).

Response: Thanks & done (line 293-294).

Line 258. For clarity insert “heterologous” or “fortuitous” before “promoter” if this is what the authors think occurred.

Response: Thanks & done (line 306).

Line 279. Change “required” to “necessary”

Response: Thanks & done (line 335).

Line 283-6. Run on sentence, add commas and/or break into two sentences.

Response: Thanks & done (line 339-342).

Line 993. Capitalize “table S3.”

Response: Thanks & done.

Reviewers' Comments:

Reviewer #1:

Remarks to the Author:

In their revised manuscript, the authors have satisfactorily addressed my concerns and present an overall convincing case for the discovery of a TBDT functioning as a protein secretion channel, even though it remains unclear how this could work. One issue that puzzled me is the apparently very different results for Oar expression reported in Fig. 7a and Sup. Fig. 7a. In the main text figure Oar is extremely abundant but in the supplementary figure it is a very minor band. Why?

Reviewer #2:

Remarks to the Author:

I have to commend the authors for continuing to put a great deal of effort into this study and for performing additional experiments to address some of the concerns I raised in my initial review. To be honest, though, after reading the revised manuscript and the authors' reply I am less convinced that they have discovered a novel secretion pathway than I was originally. As I suggested, the authors performed a chloramphenicol chase (Supp. Fig. 1) to test the idea that PopC is secreted through a periplasmic intermediate and translated "arbitrary units" into percent protein secreted (Fig. 4 legend). Based on the new information, what I now see is a phenomenon in which a very small fraction (<5%) of a periplasmic protein ends up in the supernatant over a very long period of time (hours). Perhaps it is just my own bias and my familiarity with other bacterial secretion systems in which most or all of a secreted protein ends up in the extracellular environment in a matter of minutes, but in my view the authors are greatly overselling a curious phenomenon that is difficult to define. Given the striking inefficiency of PopC secretion, I can't help but wonder if by binding to Oar in the outer membrane PopC is poised to slowly leak out of the cell by an unknown mechanism that isn't really "secretion" in the traditional sense. From the title, abstract and discussion, the authors give the impression that PopC passes through the beta barrel of Oar, which serves as a transport channel, but they present no evidence to support this idea or consider other possibilities. In my opinion the interpretation of the data still needs to be vastly toned down and needs to be much more cautious.

Specific comments:

1) Fig. S1: It is nice that the authors have included a chloramphenicol chase, but unfortunately the results are highly ambiguous. The results show that the level of Pop in the periplasm goes down over time, but do not distinguish between a scenario in which the protein is secreted and an alternative scenario in which the protein is simply degraded. If the authors wish to prove that the protein is secreted, then they need to show accumulation of the protein in the supernatant, as I originally suggested.

2) Fig. 2: I think that the authors have done a nice job of improving the presentation of the data, but unfortunately they have not clarified the role of the PMF in the secretion of PopC. The effect of the PMF inhibitors is very modest, and it is still not clear if they see a small effect on secretion because they have not completely dissipated the PMF or if they have simply altered another aspect of cell physiology indirectly. I do not agree that "We conclude that the PMF and the pH gradient are important for PopC secretion across the OM" (lines 145-146) is valid. I think that the data need to be interpreted more cautiously.

3) The authors have not addressed my concern that the instability of the Oar(delta plug) mutant indicates an overall folding defect that might explain the strong reduction in PopC secretion (original comment 7). To the best of my knowledge, there is no evidence that an outer membrane protein must fold perfectly into its native conformation to insert into the outer membrane. If such evidence has been published, please identify the relevant paper(s). Perhaps more to the point, the

authors state that the Oar(delta plug) mutant "accumulates" in the outer membrane when, in fact, the results show just the opposite: the level of the protein goes down over time and the protein almost disappears completely at the last time point. The bottom line is that the authors need to interpret the data more cautiously.

Reviewer #3:

Remarks to the Author:

In the revised manuscript the authors have done a careful and thorough job of modifying the text and have included a substantial number of new experiments/controls. As outlined in the prior reviews the work is comprehensive and adds fundamentally new insights into how proteins can be transported across the OM in gram-negative bacteria by TBDT systems. Also, as discussed in the reviews and responses, there are a number of open questions and paradoxes this work does not resolve, but in my judgment can be addressed in future work. Such puzzles include explanations for the slow and inefficient transport of PopC across the OM from the periplasm and how PopC crosses the IM without a SP. A few suggestions to improve the manuscript are listed below.

Suppl. Fig. 8 is a nice addition that provides a global phylogenetic analysis of TBDT distribution and their overall importance. Because of this I would suggest the figure be moved to the main text. The authors may also consider adding a cartoon(s) illustrating how the different domains can be positioned in a linear representation of TBDTs.

Suppl. Fig. 8 and Suppl. Table 2 appear completely redundant and therefore I would suggest removing the latter.

Line 1257: I did not find "Supplemental Table 5." However, under "All original data" I did download a excel file with a tab labeled "Fig. 8b." I believe this table should be (also) labeled "Supplemental Table 5." Under the "Read me" tab there was more relevant information, but again this tab should be clearly labeled.

Regarding Supplemental Fig. 1C, the kinetics of PopC secretion across the OM is strikingly slow, i.e. even after 24 hr much of it is retained in the periplasm. Protein secretion is typically fast, i.e. on the order of seconds to maybe minutes. If not mentioned, this point needs to be discussed in the text.

Reviewer #1 (Remarks to the Author):

In their revised manuscript, the authors have satisfactorily addressed my concerns and present an overall convincing case for the discovery of a TBDT functioning as a protein secretion channel, even though it remains unclear how this could work. One issue that puzzled me is the apparently very different results for Oar expression reported in Fig. 7a and Sup. Fig. 7a. In the main text figure Oar is extremely abundant but in the supplementary figure it is a very minor band. Why?

Response: Thank you very much for the positive feedback.

Concerning the expression levels of PopC and Oar in *E. coli*, we observed that the total level of accumulation of the two proteins in all the three strains in the experiments in Supplementary Fig. 7a is lower than in the two strains in the experiments in Fig. 7a. Importantly, the two proteins accumulate at the same level in the three cleared cell lysates in the experiments in Supplementary Fig. 7a and under these conditions the *E. coli* WT strain supports PopC secretion while the $\Delta exbB$ and $\Delta exbD$ strains do not. Thus, we conclude that ExbD and ExbD are important for the Oar-dependent secretion of PopC.

Reviewer #2 (Remarks to the Author):

I have to commend the authors for continuing to put a great deal of effort into this study and for performing additional experiments to address some of the concerns I raised in my initial review. To be honest, though, after reading the revised manuscript and the authors' reply I am less convinced that they have discovered a novel secretion pathway than I was originally. As I suggested, the authors performed a chloramphenicol chase (Supp. Fig. 1) to test the idea that PopC is secreted through a periplasmic intermediate and translated "arbitrary units" into percent protein secreted (Fig. 4 legend). Based on the new information, what I now see is a phenomenon in which a very small fraction (<5%) of a periplasmic protein ends up in the supernatant over a very long period of time (hours). Perhaps it is just my own bias and my familiarity with other bacterial secretion systems in which most or all of a secreted protein ends up in the extracellular environment in a matter of minutes, but in my view the authors are greatly overselling a curious phenomenon that is difficult to define. Given the striking inefficiency of PopC secretion, I can't help but wonder if by binding to Oar in the outer membrane PopC is poised to slowly leak out of the cell by an unknown mechanism that isn't really "secretion" in the traditional sense. From the title, abstract and discussion, the authors give the impression that PopC passes through the beta barrel of Oar, which serves as a transport channel, but they present no evidence to support this idea or consider other possibilities. In my opinion the interpretation of the data still needs to be vastly toned down and needs to be much more cautious.

Response: We previously reported that PopC is slowly secreted (Rolbetzki et al. 2008). To emphasize this point and its relevance, we have modified the Introduction to emphasize that (i) the slow regulated accumulation of p17 (the product of p25 cleavage by PopC) is important for proper development with the formation of spore-filled fruiting bodies; (ii) the slow secretion of PopC is thought to ensure the slow accumulation of p17 (line 79-84). Along the same line, it is interesting to speculate that the Oar/Ton system could be adapted to ensure slow protein secretion as opposed to the rapid secretion performed by other protein secretion systems.

Concerning the interpretation of our data, we followed the advice of reviewer #2 and edited the text carefully to not overstate our conclusions. In particular, we take great care throughout the text to state that Oar together with a functional Ton system is essential for PopC secretion across the OM, that PopC secretion depends on Oar/Ton system, or that Oar together with a functional Ton system mediate the secretion of PopC across the OM. The only place in the text in which we speculate about the mechanism underlying the Oar-dependent PopC secretion is in the Discussion (line 464-472) in which we write: "Although our data do not allow us to rule out the possibility that PopC once bound to Oar would be "handed-over" to another secretion machinery, which would have to be conserved in the OM of both *M. xanthus* and *E. coli*, our data support a simpler scenario in which Oar together

with a functional Ton system are required and sufficient for PopC secretion across the OM, and that Oar together with a functional Ton system represent a novel system for protein secretion across the OM. Combining our functional data with the mechanistic insights from TBDTs involved in import, it is also tempting to speculate that secretion of the most likely unfolded 50.8 kDa PopC protein by Oar occurs similarly to the uptake of bacteriocins by TBDTs albeit in reverse". From this writing, we believe that it is clear that this is a model.

Specific comments:

1) Fig. S1: It is nice that the authors have included a chloramphenicol chase, but unfortunately the results are highly ambiguous. The results show that the level of Pop in the periplasm goes down over time, but do not distinguish between a scenario in which the protein is secreted and an alternative scenario in which the protein is simply degraded. If the authors wish to prove that the protein is secreted, then they need to show accumulation of the protein in the supernatant, as I originally suggested.

Response: In the revised manuscript, we have now included additional experimental data in which we follow PopC accumulation in the cell-free supernatant in the presence of chloramphenicol during starvation. These new data are described in line 133-141 in which we write that "Over the course of the experiment, the PopC level in the periplasm slowly decreased and the level in the cell-free supernatant slowly increased (Supplementary Fig. 1c); by contrast, the PopC level in the cytoplasm did not decrease (Supplementary Fig. 1c). In total, these observations confirm that PopC is slowly secreted to the cell-free supernatant and demonstrate that PopC is highly enriched in the periplasm. Moreover, they support that PopC is secreted to the extracellular milieu from the periplasm supporting that PopC secretion to the extracellular milieu occurs in a two-step mechanism".

2) Fig. 2: I think that the authors have done a nice job of improving the presentation of the data, but unfortunately they have not clarified the role of the PMF in the secretion of PopC. The effect of the PMF inhibitors is very modest, and it is still not clear if they see a small effect on secretion because they have not completely dissipated the PMF or if they have simply altered another aspect of cell physiology indirectly. I do not agree that "We conclude that the PMF and the pH gradient are important for PopC secretion across the OM" (lines 145-146) is valid. I think that the data need to be interpreted more cautiously.

Response: Thanks for pointing this out to us. We have modified the text in line 157-158 to "These data support that the PMF and the pH gradient are important for PopC secretion across the OM".

3) The authors have not addressed my concern that the instability of the Oar(delta plug) mutant indicates an overall folding defect that might explain the strong reduction in PopC secretion (original comment 7). To the best of my knowledge, there is no evidence that an outer membrane protein must fold perfectly into its native conformation to insert into the outer membrane. If such evidence has been published, please identify the relevant paper(s). Perhaps more to the point, the authors state that the Oar(delta plug) mutant "accumulates" in the outer membrane when, in fact, the results show just the opposite: the level of the protein goes down over time and the protein almost disappears completely at the last time point. The bottom line is that the authors need to interpret the data more cautiously.

Response: In the revised manuscript, we have modified the text to "These data support that the plug domain is important for Oar to support PopC secretion" (line 273-274) and "The TonB box in the plug domain mediates the interaction between the plug domain of well-characterized TBDTs and the periplasmic CTD of TonB. If the suggested importance of the Oar plug domain" (line 277-278) and "We conclude that the TonB box in the Oar plug domain is important for Oar function and PopC secretion, likely by directly interacting with the CTD of TonB1" (line 285-286).

Reviewer #3 (Remarks to the Author):

In the revised manuscript the authors have done a careful and thorough job of modifying the

text and have included a substantial number of new experiments/controls. As outlined in the prior reviews the work is comprehensive and adds fundamentally new insights into how proteins can be transported across the OM in gram-negative bacteria by TBDT systems. Also, as discussed in the reviews and responses, there are a number of open questions and paradoxes this work does not resolve, but in my judgment can be addressed in future work. Such puzzles include explanations for the slow and inefficient transport of PopC across the OM from the periplasm and how PopC crosses the IM without a SP. A few suggestions to improve the manuscript are listed below.

Response: Thank you very much.

Suppl. Fig. 8 is a nice addition that provides a global phylogenetic analysis of TBDT distribution and their overall importance. Because of this I would suggest the figure be moved to the main text. The authors may also consider adding a cartoon(s) illustrating how the different domains can be positioned in a linear representation of TBDTs.

Response: We followed the advice of the reviewer and moved the figure to the main text as Fig. 9. We have not added a cartoon of the domain structure in a linear representation because we believe that the domain architecture is already evident from the upper part of Fig. 9.

Suppl. Fig. 8 and Suppl. Table 2 appear completely redundant and therefore I would suggest removing the latter.

Response: We followed the suggestion of the reviewer and deleted Suppl. Table 2.

Line 1257: I did not find “Supplemental Table 5.” However, under “All original data” I did download a excel file with a tab labeled “Fig. 8b.” I believe this table should be (also) labeled “Supplemental Table 5.” Under the “Read me” tab there was more relevant information, but again this tab should be clearly labeled.

Response: We are sorry about the confusion. We have now included all source data in the Source Data file.

Regarding Supplemental Fig. 1C, the kinetics of PopC secretion across the OM is strikingly slow, i.e. even after 24 hr much of it is retained in the periplasm. Protein secretion is typically fast, i.e. on the order of seconds to maybe minutes. If not mentioned, this point needs to be discussed in the text.

Response: We previously reported that PopC is slowly secreted (Rolbetzki et al. 2008). To emphasize this point and its relevance, we have modified the Introduction to emphasize that (i) the slow regulated accumulation of p17 (the product of p25 cleavage by PopC) is important for proper development with the formation of spore-filled fruiting bodies; (ii) the slow secretion of PopC is thought to ensure the slow accumulation of p17 (line 79-84).

Reviewers' Comments:

Reviewer #2:

Remarks to the Author:

The authors have made some improvements in their revised manuscript, but several of their main conclusions are still not convincingly supported by the data. To their credit, the authors now emphasize that PopC was previously shown to be secreted slowly and link the slow secretion to the regulated production of p17, but they have not addressed my concern about the inefficiency of PopC secretion. Is 5% of the protein secreted because only a few cells participate in the secretion reaction or because the protein "leaks" out by some non-conventional mechanism? Furthermore, the authors still overinterpret the data by stating that the Oar/Ton system "mediates" secretion. At best they might argue that Oar/Ton is required for secretion, i.e., it must be present for PopC to be secreted. The word "mediates", however, suggests that Oar/Ton is part of the secretion apparatus itself and conjures up the unsupported notion that PopC passes through the pore of the Oar protein. Finally, I would note that the authors only partially addressed one of my previous "specific comments" and did not address the other two.

Specific comment 1: The new panel adds a bit more information to the chloramphenicol chase experiment (Fig. S1), but does not support the conclusion that the "level [of PopC] in the cell-free supernatant slowly increased" (lines 135-136). The level of secreted protein only rises at the last (24 h) time point. I can imagine that after 24 h in chloramphenicol the cells started to lyse and release PopC, but this possibility would need to be addressed by performing a control experiment that evaluates cell integrity. The results of this experiment are still very shaky. The experiment needs to be repeated with proper controls.

Specific comment 2: The change that the authors have made does not address my concern. The effect of the PMF inhibitors is modest, and the authors still cannot determine whether the effect on secretion is due to an incomplete dissipation of the PMF or to an alteration of another aspect of cell physiology (i.e., to an indirect effect). The conclusion that the PMF is "important" for PopC secretion is not clearly supported by the data.

Specific comment 3: The Oar(delta plug) variant does not "accumulate at a lower level than the WT protein", as the authors state (line 270). On the contrary, the variant is degraded over time. This observation strongly suggests that the variant does not fold correctly. For this reason the role of the plug domain cannot be evaluated, and the role of the plug domain in secretion remains unclear.

Reviewer #2 (Remarks to the Author):

The authors have made some improvements in their revised manuscript, but several of their main conclusions are still not convincingly supported by the data. To their credit, the authors now emphasize that PopC was previously shown to be secreted slowly and link the slow secretion to the regulated production of p17, but they have not addressed my concern about the inefficiency of PopC secretion. Is 5% of the protein secreted because only a few cells participate in the secretion reaction or because the protein “leaks” out by some non-conventional mechanism? Furthermore, the authors still overinterpret the data by stating that the Oar/Ton system “mediates” secretion. At best they might argue that Oar/Ton is required for secretion, i.e., it must be present for PopC to be secreted. The word “mediates”, however, suggests that Oar/Ton is part of the secretion apparatus itself and conjures up the unsupported notion that PopC passes through the pore of the Oar protein. Finally, I would note that the authors only partially addressed one of my previous “specific comments” and did not address the other two.

Response: We followed the advice of the reviewer and revised the text and title to avoid using “mediate” and changed the title to “A TonB-dependent transporter is required for secretion of the protease PopC across the bacterial outer membrane”. In addition, we modified the text throughout to avoid overstatements. In particular, we also added in line 470-472 “It is important to emphasize that it has not been demonstrated that PopC passes through the Oar β -barrel” to make clear that we have not demonstrated how precisely PopC crosses the OM.

We have shown that starving *M. xanthus* cells secrete PopC, i.e. cytoplasmic and periplasmic control proteins are not detected in the cell-free supernatant (ref. 7 and here). We report here that secretion of PopC in *M. xanthus* as well as in *E. coli* depends on the TBDT Oar. Moreover, in *E. coli*, we demonstrate that PopC secretion depends on an intact Ton system. We do not see how these observations can be reconciled with a “leaking” process.

We have also shown that PopC is secreted slowly (ref. 7, 8 & here) and approx. 5% of PopC is secreted. We do not know the mechanism that underlies the slow secretion of PopC. Similarly, we do not know why only 5% of PopC is secreted, i.e. are only some of the cells secreting? These questions are important to address; however, we strongly feel – and hope that you agree with us – that addressing these questions go beyond the scope of this manuscript. We have added in line 481-482 “Also, the mechanism underlying the slow PopC secretion kinetics warrants further analyses”. It is of course interesting to speculate that the Oar/Ton system could be adapted to ensure slow protein secretion as opposed to the rapid secretion performed by other protein secretion systems.

Specific comment 1: The new panel adds a bit more information to the chloramphenicol chase experiment (Fig. S1), but does not support the conclusion that the “level [of PopC] in the cell-free supernatant slowly increased” (lines 135-136). The level of secreted protein only rises at the last (24 h) time point. I can imagine that after 24 h in chloramphenicol the cells started to lyse and release PopC, but this possibility would need to be addressed by performing a control experiment that evaluates cell integrity. The results of this experiment are still very shaky. The experiment needs to be repeated with proper controls.

Response: We apologize for not being clearer. We have added in line 128-133, that we previously demonstrated (ref. 8) that the slow PopC secretion kinetics displayed by starving cells treated with chloramphenicol is indistinguishable from the slow PopC secretion kinetics of untreated starving cells. Because untreated cells secrete PopC, i.e. cytoplasmic and periplasmic

control proteins are not detected in the cell-free supernatant (here and ref. 7), we infer that cells are also secreting PopC in the presence of chloramphenicol and not simply “leaking” PopC. Based on this information, we do not believe that additional control experiments are needed.

Specifically, we write that in line 128-142 “We previously showed that starving cells, in the presence of the translation inhibitor chloramphenicol, secrete PopC to the extracellular milieu following the same kinetics as in untreated starving cells during the first 24 h of starvation, i.e. the PopC level in total cell extracts slowly decreases while the level in the cell-free supernatant slowly increases (ref. 8). Thus, it was concluded that PopC synthesis and secretion are not coupled and that preformed PopC is secreted to the extracellular milieu (ref. 8). Therefore, to determine whether PopC is secreted to the extracellular milieu from the cytoplasm or the periplasm, we examined the level of PopC accumulation in fractions enriched for periplasmic and cytoplasmic proteins as well as in the cell-free supernatant in starving cells treated with chloramphenicol. Over the course of the experiment, the PopC level in the periplasm slowly decreased while the level in the cytoplasm did not decrease (Supplementary Fig. 1c). Moreover, and as previously observed (ref. 8), the PopC level in the cell-free supernatant increased (Supplementary Fig. 1c). In total, these observations demonstrate that PopC is highly enriched in the periplasm. Moreover, they support that PopC is secreted to the extracellular milieu from the periplasm supporting that PopC secretion to the extracellular milieu occurs in a two-step mechanism”.

Specific comment 2: The change that the authors have made does not address my concern. The effect of the PMF inhibitors is modest, and the authors still cannot determine whether the effect on secretion is due to an incomplete dissipation of the PMF or to an alteration of another aspect of cell physiology (i.e., to an indirect effect). The conclusion that the PMF is “important” for PopC secretion is not clearly supported by the data.

Response: We followed the advice of the reviewer and (1) modified the Abstract to say that “proton motive force has a role in ...” (2) removed mentioning of PMF in the last para of the Introduction, (3) in line 144, changed the headline of the section to “PMF has a role in PopC secretion”, (4) conclude in line 158-160 “While it cannot be excluded that CCCP and nigericin indirectly affect PopC secretion, these data support the notion that the PMF and the pH gradient might have a role in PopC secretion across the OM” (5) in line 168 changed the text to write that “...by a novel mechanism that might involve the PMF”, (6) in line 426 changed the text to write that “...PMF has a role in PopC secretion....”.

Specific comment 3: The Oar(delta plug) variant does not “accumulate at a lower level than the WT protein”, as the authors state (line 270). On the contrary, the variant is degraded over time. This observation strongly suggests that the variant does not fold correctly. For this reason the role of the plug domain cannot be evaluated, and the role of the plug domain in secretion remains unclear.

Response: We followed the advice of the reviewer and write in line 271-274 “The Oar variant that lacked the plug domain accumulated at a lower level than the WT protein in total cell extracts and appeared to be degraded during starvation, but was still present in the OM (Fig. 5c, d). This variant did not support PopC secretion to the extracellular milieu.”

And, in line 278-288 “To try to further clarify whether the Oar plug domain has a function in PopC secretion, we focused on the conserved TonB box in the plug domain. In well-characterized TBBDTs, the TonB box mediates the interaction between the plug domain and the periplasmic CTD of TonB. Therefore, if the Oar plug domain is important for PopC secretion, then the prediction is that the TonB box in the Oar plug is also important for PopC secretion”. In line 287-288 we conclude that “...the TonB box in the Oar plug domain is important for Oar function in PopC secretion, likely by directly interacting with the CTD of TonB1”. Finally, in line 446-446 we write that “.., our data support that the TonB box in the plug domain is important for PopC secretion.....”.